# FlowDreamer: Exploring High Fidelity Text-to-3D Generation Via Rectified Flow

## Abstract

Recent advances in text-to-3D generation have made significant progress. In particular, with the pretrained diffusion models, existing methods predominantly use Score Distillation Sampling (SDS) to train 3D models such as Neural Radiance Fields (NeRF) and 3D Gaussian Splatting (3D GS). However, a hurdle is that they often encounter difficulties with over-smoothing textures and over-saturating colors. The rectified flow model – which utilizes a simple ordinary differential equation (ODE) to represent a straight trajectory – shows promise as an alternative prior to text-to-3D generation. It learns a *time-independent* vector field, thereby reducing the ambiguity in 3D model update gradients that are calculated using *time-dependent* scores in the SDS framework. In light of this, we first develop a mathematical analysis to seamlessly integrate SDS with rectified flow model, paving the way for our initial framework known as Vector Field Distillation Sampling (VFDS). However, empirical findings indicate that VFDS results in over-smoothing outcomes. Therefore, we analyze the grounding reasons for such a failure from the perspective of ODE trajectories. On top, we propose a novel framework, named **FlowDreamer**, which yields high-fidelity results with richer textual details and faster convergence. The key insight is to leverage the coupling and reversible properties of the rectified flow model to search for the corresponding noise, rather than using randomly sampled noise as in VFDS. Accordingly, we introduce a novel Unique Couple Matching (**UCM**) loss, which guides the 3D model to optimize along the same trajectory. Our FlowDreamer is superior in its flexibility to be applied to both NeRF and 3D GS. Moreover, we highlight the intriguing open questions, such as initialization challenges in NeRF and sampling techniques, to benefit the research community.

## 1 Introduction

3D generation enjoys broad applications in diverse fields, such as the Metaverse, games, education, architecture design, and films, and has attracted significant research interest recently (Xie et al., 2024; Wang et al., 2024b; Tang et al., 2024; Poole et al., 2022; Chen et al., 2023a; Lin et al., 2023; Jain et al., 2022; Tang et al., 2023; Jiang et al., 2024). Text-to-3D generation – which generates 3D contents from user-input text – has emerged as one of the promising 3D generation paradigms due to its ease of use (Wang et al., 2023; Yi et al., 2023; Wang et al., 2024a; Nichol et al., 2022; Jun & Nichol, 2023).

Recently, with the advances of text-to-2D image synthesis techniques based on the diffusion models, text-to-3D generation also undergoes a surge of research interest. A seminal work, Dream-Fusion (Poole et al., 2022) sets a cornerstone by proposing Score Distillation Sampling (SDS) to address this issue by leveraging pretrained text-to-image diffusion model, to train Neural Radiance Fields (NeRF) (Mildenhall et al., 2021). It has been rapidly evolved to 3D Gaussian Splatting (3D GS) (Kerbl et al., 2023; Tang et al., 2023; Yi et al., 2023) for faster training and rendering.

Despite the success, existing works (Lin et al., 2023; Poole et al., 2022; Zhu et al., 2023) unveil that SDS suffers from issues such as over-smoothing textures and over-saturating colors. For this reason, some attempts, *e.g.*,Wang et al. (2024a); Liang et al. (2023); Wu et al. (2024) improve SDS from different perspectives. For example, ProlificDreamer (Wang et al., 2024a) introduces variational score distillation (VSD), which models 3D parameters as random variables to distill 3D assets. However, it requires significantly more time to optimize. Consistent3D (Wu et al., 2024)

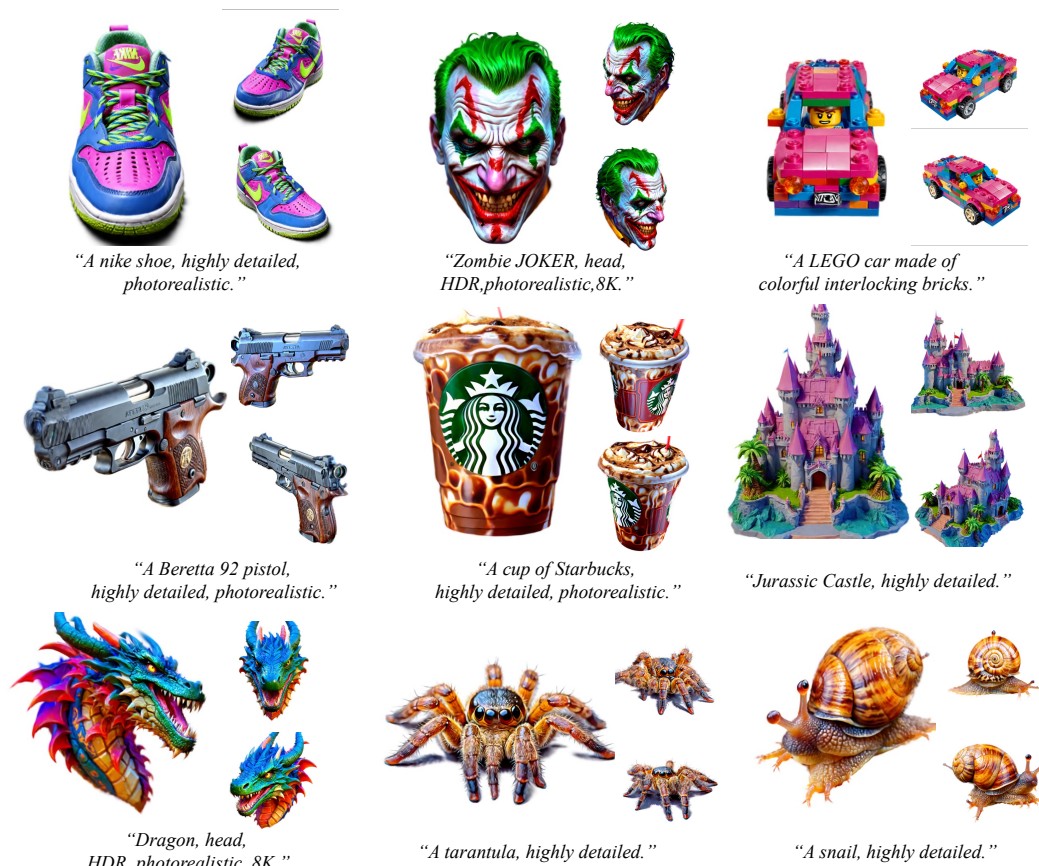

*"A nike shoe, highly detailed, photorealistic."*    *"Zombie JOKER, head, HDR,photorealistic,8K."*    *"A LEGO car made of colorful interlocking bricks."*

*"A Beretta 92 pistol, highly detailed, photorealistic."*    *"A cup of Starbucks, highly detailed, photorealistic."*    *"Jurassic Castle, highly detailed."*

*"Dragon, head, HDR, photorealistic, 8K."*    *"A tarantula, highly detailed."*    *"A snail, highly detailed."*

Figure 1: FlowDreamer uses a pretrained rectified flow model to generate high-fidelity results from text prompts. It can generate not only highly realistic objects, such as guns and shoes, but also fantastical ones, such as dragon heads.

designs a consistency distillation sampling method to train the 3D model. Nevertheless, the quality improvements are still limited. LucidDreamer (Liang et al., 2023) proposes interval score matching (ISM) loss in the diffusing trajectory, but the loss is formulated based on strong assumptions and drops many terms with the same scale.

Recently, flow matching approaches (Liu et al., 2022; Lipman et al., 2022) pave new ways for fast and high-quality generation. Among them, rectified flow model (Liu et al., 2022; Esser et al., 2024; Liu et al., 2023) uses a simple ordinary differential equation (ODE) to represent a straight trajectory. It learns a *time-independent* vector field, but Liu et al. (2022); Lipman et al. (2022) indicates that the trajectory is not completely straight. Whereas (Liu et al., 2022) points out that it is still straighter than curved diffusion trajectories. Thereby rectified flow can reduce the ambiguity in 3D model update gradients Whereas score (Song et al., 2020b) is *time-dependent*, meaning that SDS optimizing over uniformly sampled values of $t$ can produce different gradient directions. Owning to its merits, we interestingly find that it could serve as an alternative prior for text-to-3D generation.

In light of this, we first develop a mathematical analysis to seamlessly integrate SDS with rectified flow model. This enables us to build up an initial framework, named as Vector Field Distillation Sampling (VFDS). However, empirical results demonstrate that VFDS leads to over-smoothing textures (See Figure 2). To this end, we further analyze the grounding reasons for such a failure from the perspective of ODE tra-

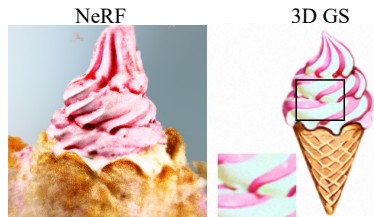

NeRF          3D GS

*"a DSLR photo of an ice cream."*

Figure 2: An example of over-smoothing results.

jectories. This way, we find that, *as VFDS randomly samples noise, it leads to multiple ODE trajectories in nearly the same images*, *i.e.*, from camera poses with mild differences (See Figure 5). This causes inconsistent update directions, leading to over-smoothing textural issues.

Buttressed by the analysis, we propose a novel framework, named **FlowDreamer**, which yields high-fidelity results with rich textual details. *The key idea of FlowDreamer is to leverage the coupling and reversible properties of rectified flow model.* Importantly, the reversible property is explored to search the corresponding noise from the image while the coupling property ensures that the corresponding noise is unique. For formality, we define it as a *push-backward* process to avoid the aforementioned over-smoothing issue caused by multiple ODE trajectories and better make our update directions consistent. Empirical experiments show that the *push-backward* process is efficient as three Euler discretization steps are sufficient for it to achieve plausible performance, see Figure 1.Accordingly, we propose a novel Unique Couple Matching (**UCM**) loss that guides the 3D model to learn the same trajectory. Our FlowDreamer also enjoys high flexibility as it can be applied to either NeRF or 3D GS generation settings.

We conduct extensive experiments under diverse generation settings, demonstrating high-fidelity results with rich details of FlowDreamer, as shown in Figure 1. When exploring the application of FlowDreamer to NeRF, we observe some interesting phenomena. Moreover, we identify some intriguing open questions. First is the initialization problem that emerges when applying FlowDreamer to NeRF. This is because the distribution of the initialized image from NeRF is undefined in the Rectified flow diffusion space. We use a warm-up strategy to mitigate this issue. Secondly, the *push-backward* process with different numbers of function evaluations (NFE) and sampling methods can generate some interesting results.

In summary, our major contributions are as follows:

- We are the *first* to explore a new direction by leveraging the rectified flow model as an alternative prior for text-to-3D generation. We accordingly build a mathematical analysis to adapt SDS to rectified flow model, paving the way for an initial framework – VFDS. Empirical results demonstrate that VFDS still leads to over-smoothing. We further analyze the underlying reasons for this issue from the perspective of ODE trajectories.

- Based on the analysis, we further propose a text-to-3D framework, FlowDreamer, with a novel UCM loss. The loss is build upon the *push-backward* process to search for corresponding noise, rather than using randomly sampled noise in VFDS.

- Extensive experiments in both NeRF and 3D GS generation settings demonstrate high-fidelity results with rich details for our FlowDreamer. We also identify some interesting open questions, such as initialization issues for NeRF and sampling techniques in the noise search process.

## 2 RELATED WORKS

**Text-to-3D generation.** It aims to create 3D assets from user-input text. DreamFusion (Poole et al., 2022) proposes Score Distillation Sampling (SDS) that leverages the pretrained diffusion models to train a NeRF. However, SDS suffers from issues such as over-smoothing textures, low resolution, slow convergence, multi-faced problem (Wang et al., 2024a; Lin et al., 2023; Poole et al., 2022), *etc*. Magic3D (Lin et al., 2023) designs a coarse-to-fine two-stage training pipeline and changes the 3D model to DMtet (Shen et al., 2021) to improve the resolution of the generated 3D results. Later on, some works (Tang et al., 2023; Yi et al., 2023; Liang et al., 2023; Chen et al., 2023b; Jiang et al., 2024; Li et al., 2024; Jiang & Wang, 2024) take 3D GS as the 3D model for faster training and rendering. Recently, to overcome the multi-face problem, some works (Shi et al., 2023; Wang & Shi, 2023; Tang et al., 2024) fine-tune the pretrained diffusion models to generate multi-view images.

**Design variant of SDS loss.** To overcome the issue of over-smoothing textural issues, some attempts (Wang et al., 2023; 2024a; Liang et al., 2023; Wu et al., 2024; Zhu et al., 2023; Katzir et al., 2023; Yu et al., 2023) focus on designing different SDS losses. For example, Wang et al. (2023) proposes Score Jacobian Chaining, which applies the chain rule to the estimated score to enhance generation quality. ProlificDreamer (Wang et al., 2024a) proposes VSD to model 3D parameters as

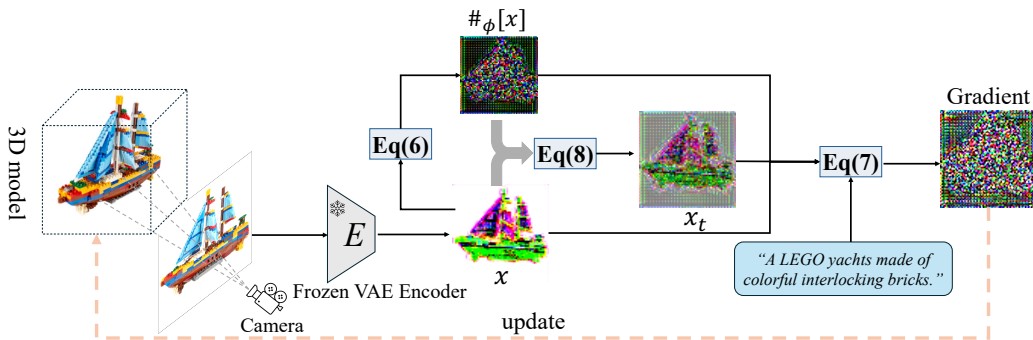

Figure 3: **Illustration of our FlowDreamer**. Images of random views from different camera poses are sampled and then input to the VAE encoder to obtain the latents. We replace the randomly sampled noise $\epsilon$ in VFDS with $\#_\phi[x]$ via the *push-backward* process. Next, we sample $t$ from $U[0, 1]$ and interpolate to obtain $x_t$. Finally, the UCM loss with the conditional prompt is applied to update the 3D model.

random variables to distill 3D assets. Although with improved quality, it requires a much longer time to optimize. Consistent3D (Wu et al., 2024) designs a consistency distillation sampling method to train the 3D model. Nevertheless, the quality improvements are limited. LucidDreamer (Liang et al., 2023) proposes ISM loss in the diffusing trajectory, but the loss drops many terms with the same scale. These methods build loss towards either DDPM (Ho et al., 2020) or DDIM (Song et al., 2020a) models. *By contrast, we propose a novel UCM loss that is build upon the push-backward process to search for corresponding noise, rather than using randomly sampled noise with rectified flow-based models. Our UCM demonstrates high-fidelity results with rich details and faster convergence under either NeRF or GS generation settings.*

**Diffusion and Flow based models.** Recent advances in text-to-image generation have witnessed a significant progress. Diffusion models define a process, which progressively converts a distribution of training data to pure Gaussian noise. By learning the reverse process, one can sample data following the distribution. DDPM (Ho et al., 2020) employs a Markov chain to achieve the above process, while DDIM (Song et al., 2020a) proposes to reduce the iteration step by maintaining the marginal distribution hence free from the restriction of Markov property. The diffusion process can be essentially modeled as stochastic differential equations (SDE) (Song et al., 2020b) or ordinary stochastic equations (ODE) (Lipman et al., 2022). On the other hand, flow-base models (Liu et al., 2022; Lipman et al., 2022; Liu et al., 2023) pave new ways for faster and higher-quality generation.

Recently, rectified flow (Liu et al., 2022) – one of the ODE methods – defines a simple process, optimizing the trajectories in diffusion space to be as straight as possible. *We are the first to explore a new direction by leveraging the rectified flow model as an alternative prior for text-to-3D generation. We propose a novel UCM loss, built upon the push-backward process to search for corresponding noise, Extensive experiments in both NeRF and 3D GS generation settings demonstrate higher-fidelity results with richer details and faster convergence.*

## 3 FLOWDREAMER

**Overview.** An overview of our proposed FlowDreamer is depicted in Figure 3. The key insight is to leverage the coupling and reversible properties of the rectified flow model to search for the corresponding noise, rather than using randomly sampled noise as in our initial framework VFDS (see Sec. 3.1). Accordingly, we introduce a novel Unique Couple Matching (**UCM**) loss in Sec. 3.2, which guides the 3D model to optimize along the same trajectory. Finally, our FlowDreamer can be applied to two types of 3D models: 3D Gaussian splatting (Kerbl et al., 2023) and NeRF (Mildenhall et al., 2021) settings. Now let's describe the details (see Sec. 3.3).

### 3.1 VFDS: SDS IN THE LENS OF RECTIFIED FLOW

**Adapting SDS to the rectified flow framework.** We first briefly introduce the rectified flow (Liu et al., 2022). Let $\pi_1$ and $\pi_0$ denote Gaussian distribution $\mathcal{N}(\mathbf{0}, \mathbf{I})$ and data distribution, respectively.

Figure 4: (**a**): **An illustration of the reversible and coupling properties of the rectified flow model**. The reversible property indicates that $\epsilon$ can map to $x_0$, and $x_0$ can map to $\epsilon$ by reversing the direction of $v_\phi$. The coupling property indicates that $\epsilon$ and $x_0$ can only form a unique coupling. For example, $\epsilon_2$ and $x_0^2$ form a coupling $(\epsilon_2, x_0^2)$; therefore, $\epsilon_2$ and $x_0^1$ can't form a coupling $(\epsilon_2, x_0^1)$ again. (**b**): **An illustration of the trajectories of diffusion and rectified flow.** The gradient direction of the diffusion trajectory varies with different $t$, while the rectified flow roughly remains the same for different $t$ under ideal circumstances.(**Please note: The rectified flow trajectory is actually not completely straight; this is just an idealized illustration.**)

$\epsilon$ and $x_0$ are respectively sampled from $\pi_1$ and $\pi_0$. Rectified flow defines the forward process as(to simplify the representation, below, $x_0$ and $x_t$ indicate the latent space.):

$$x_t = t\epsilon + (1 - t)x_0, t \in [0, 1] \tag{1}$$

Accordingly, the reverse process follows the Ordinary Differential Equation (ODE) to map $\epsilon$ to $x_0$.

$$dx_t = v_\phi(x_t, t)dt, t \in [0, 1], \tag{2}$$

where the velocity $v_\phi$ is estimated by a learnable network $\phi$. The model is trained as follows:

$$\mathcal{L}_{\text{rflow}}(\phi, \mathbf{x}) = \mathbb{E}_{x_0 \sim p_0, \epsilon \sim \mathcal{N}(\mathbf{0}, \mathbf{I}), t \sim U[0,1]} \left[ w(t) \|(\epsilon - x_0) - v_\phi(x_t, t)\|_2^2 \right], \tag{3}$$

where $w(t)$ is a time-dependent weighting function, $U[0, 1]$ denotes the uniform distribution within $[0, 1]$. Because the rectified flow model is an ODE model, it has reversible and coupling properties. The $\epsilon$ from the Gaussian noise distribution is uniquely coupled with the $x$ from the data distribution. Moreover, the rectified flow is reversible, as shown in Figure 4(a). Specifically,

**1) Reversible property**: The $\epsilon$ from the Gaussian noise distribution can map to $x$, while $x$ from the data distribution can also map to $\epsilon$.

**2) Coupling property**: The $\epsilon$ is determined, and the $x$ generated by the same rectified flow model is unique. Conversely, the generated $\epsilon$ is unique for a given $x$.

Now, we elucidate how to adapt SDS to the rectified flow to build an initial framework, called *Vector Field Distillation Sampling* (VFDS). The loss, denoted as $\mathcal{L}_{\text{VFDS}}$, can be written as:

$$\nabla_\theta \mathcal{L}_{\text{VFDS}}(\phi, x = g(\theta, \mathbf{c})) = \mathbb{E}_{\epsilon, t} \left[ w(t) \left( v_\phi(x_t, t) - (\epsilon - x) \right) \left( \underbrace{\frac{\partial v_\phi(x_t, t)}{\partial x_t} \frac{\partial x_t}{\partial x}}_{\text{transformer Jacobian}} + 1 \right) \frac{\partial x}{\partial \theta} \right] \tag{4}$$

where $\theta$ is the 3D model parameters, $x = g(\theta, \mathbf{c})$ denotes a rendered image from a camera pose $\mathbf{c}$, $\epsilon$ denotes randomly sampled Gaussian noise, $t \sim U[0, 1]$. Following the convention of the SDS, we omit the transformer Jacobian term for effective training. Therefore, $\left( \frac{\partial v_\phi(x_t, t)}{\partial x_t} \frac{\partial x_t}{\partial x} + 1 \right)$ becomes a constant and can be absorbed by $w(t)$, so we have

$$\nabla_\theta \mathcal{L}_{\text{VFDS}}(\phi, \mathbf{x} = g(\theta, \mathbf{c})) \triangleq \mathbb{E}_{\epsilon, t} \left[ w(t) \left( v_\phi(x_t, t) - (\epsilon - x) \right) \frac{\partial x}{\partial \theta} \right] \tag{5}$$

Based on Equation 5, we train a 3D model utilizing a pretrained rectified flow model. The diffusion model's trajectory is (Lipman et al., 2022), and the score (Song et al., 2020b; Poole et al., 2022) direction varies with different $t$. (see Figure 4(b)). We denote $\epsilon_\phi(x_t, t)$ as the score function, where $\phi$ represents the parameters of the denoise network, and $x_t$ follows the diffusion forward process. The

trajectory of the rectified flow model (Liu et al., 2022; 2023) is straighter than curved diffusion trajectories, and the vector field direction, $v_\phi(x_t, t)$ is more consistent under different $t$ compared to the score direction, $\epsilon_\phi(x_t, t)$. In our VFDS framework – where $t \sim U[0,1]$ is used in every optimization step – the VFDS optimization direction is more consistent. However, the over-smoothing issue of SDS still exists. Therefore, we further analyze the grounding reasons for the over-smoothing issue from the perspective of ODE trajectories. **Elucidating SDS Over-smoothing Issue with VFDS.**

Because rectified flow is an ODE (Lipman et al., 2022) model, it has the coupling property. Now, we analyze the term $(v_\phi(x_t, t) - (\epsilon - x))$ in Equation 5, where $x_t = t\epsilon + (1 - t)x$, $\epsilon \sim \mathcal{N}(\mathbf{0}, \mathbf{I})$ is a random sampled noise and $x$ is the generated image from 3D model. As shown in Figure 5, VFDS randomly samples noise $\epsilon$, which leads to multiple ODE trajectories in the same image $x$.

When camera poses have only mild differences, the rendered images appear nearly identical. Different ODE trajectories cause inconsistent update directions, which means that directions of $(v_\phi(x_t, t) - (\epsilon - x))$ are inconsistent. Figure 5 illustrates a toy example. $\epsilon_1, \epsilon_2, \epsilon_3$ are noise randomly sampled from $\mathcal{N}(\mathbf{0}, \mathbf{I})$ independently. According to Equation 1, we can get $x_t^1, x_t^2, x_t^3$, which looks like a blurred image (a hamburger with noise). Following the above analysis, the same $x$ (rightmost hamburger in Figure 5) together with different $\epsilon_1, \epsilon_2, \epsilon_3$ form different trajec-

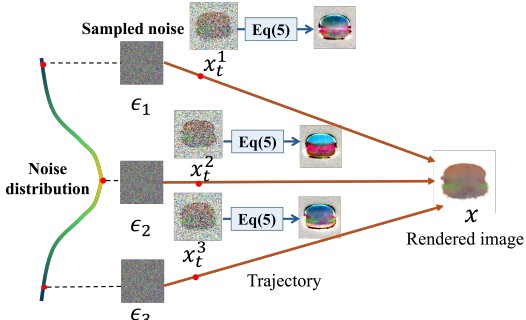

Figure 5: Illustration for over-smoothing analysis. An image is coupled with multiple randomly sampled noises, causing the 3D model to learn ODE trajectories.

tories. During the VFDS training process, the rectified flow model takes $x_t^1, x_t^2, x_t^3$ as input and outputs the estimation of trajectory gradients, as shown in Figure 5. Note that the fitting targets are $\epsilon_1 - x, \epsilon_2 - x, \epsilon_3 - x$ respectively. The 3D model finally learns the multiple trajectories gradient, causing the over-smoothing issue.

## 3.2 UNIQUE COUPLE MATCHING LOSS

In Sec. 3.1, we have identified the grounding reason for the over-smoothing issue, It is caused by optimizing multiple trajectories during the VFDS training. On top, we propose a novel Unique Couple Matching (UCM) loss that guides the 3D model to optimize in the same trajectory. *Our key idea is to leverage the coupling and reversible properties of rectified flow model.*

We define the process from $x$ to $\epsilon$ as *push-backward* process, denoted $\#$, which can be written as:

$$\#_\phi[x] = x + v_\phi(x, \delta_{T_0})\Delta_{T_1} + v_\phi(x_{\delta_{T_1}}, \delta_{T_1})\Delta_{T_2} + \cdots + v_\phi(x_{\delta_{T_{n-1}}}, \delta_{T_{n-1}})\Delta_{T_n}$$
$$x_{\delta_{T_{n-1}}} = x_{\delta_{T_{n-2}}} + v_\phi(x_{\delta_{T_{n-2}}}, \delta_{T_{n-2}})\Delta_{T_{n-1}}, n \geq 2 \tag{6}$$

where, $\sum_{i=1}^n \Delta_{T_i} = 1$, $\delta_{T_0} = 0$. And $\#_\phi[x]$ denotes iteratively calculate $v_\phi(x_{\delta_{T_i}}, t)$ to backtrack to the $\epsilon$ from $x$ in Equation 6. Due to the reversible property of rectified flow, we can search for a noise $\epsilon$ from $x$ in the VFDS framework. Additionally, because of the aforementioned coupling property, the search noise is unique. By replacing randomly sampled noise $\epsilon$ to $\#_\phi[x]$ in Equation 5, our UCM loss is defined as follows:

$$\nabla_\theta \mathcal{L}_{\text{UCM}}(\theta, \mathbf{x} = g(\theta, \mathbf{c})) \triangleq \mathbb{E}_t \left[ w(t) \left( v_\phi(x_t, t) - (\#_\phi[x] - x) \right) \frac{\partial x}{\partial \theta} \right] \tag{7}$$

$$where, x_t = t\#_\phi[x] + (1 - t)x \tag{8}$$

As shown in Figure 3 we can use *push-backward* instead of randomly sampled noise in VFDS, which guides the 3D model to optimize in the same trajectory.

### 3.3 FlowDreamer for NeRF and 3D GS

Based on the UCM loss, we propose a novel FlowDreamer framework. It yields high-fidelity results with richer textual details. FlowDreamer can be applied to two types of 3D models: 3D Gaussian splatting (Kerbl et al., 2023) and NeRF (Mildenhall et al., 2021).

**Application to 3D Gaussian Splatting.** We use points generated by the text-to-3D generator (Nichol et al., 2022) as parameter initialization. Then, we directly train the 3D GS model using our UCM loss. Our FlowDreamer yields high-fidelity results with richer textual details (see Figure 1).

**Application to NeRF and the initialization issue.** For NeRF, however, a direct application does not perform well. We call this the *initialization issue* of NeRF, as we will explain the reasons below. When searching the noise $\epsilon$ for a given image $x$, *i.e.*, *push-backward* process, we use the rectified flow model $v_\phi$. It defines a mapping from data distribution $\pi_0$ to noise distribution $\pi_1$, where $\pi_0$ is the distribution of its pre-training datasets. The effectiveness of the *push-backward* process depends on that the input distribution for $v_\phi$ should be aligned with or at least approximate to $\pi_0$ or $\pi_1$. Otherwise, the input lies in an undefined area for $v_\phi$ hence the output is unreasonable.

When training NeRF from scratch, it can hardly generate reasonable images based on its randomly initialized parameters. Therefore, the input distribution (denoted as $\pi^{nf}$) is far from $\pi_0$ and $\pi_1$, causing the rectified flow model $v_\phi$ difficult to estimate the gradient of the ODE trajectory. To solve this issue, we temporarily use the naive VFDS training to warm up as a remedy. We view this initialization issue as an open question and advocate further investigations.

As for 3D GS models, the issue does not exist. The example can be found in the supplementary material, when the prompt "*A English cottage with stone walls*" is provided, the NeRF Initialization is simply a gray image, while the output of the 3D GS model (initialized by Point-E (Nichol et al., 2022)) resembles a cottage. This indicates the initial distribution of the 3D Gaussian splatting model is more approximate to $\pi_0$. The result of *push-backward* process is also more effective, as they are closer to the gradient from images output by the trained model.

## 4 Experiments

### 4.1 3D Generation Settings

**3D Gaussian Splatting Generation.** We compare our FlowDreamer with DreamGaussian (Tang et al., 2023), GaussianDreamer (Yi et al., 2023) and LucidDreamer (Liang et al., 2023). These 3D GS SoTA baselines are based on their official code by employing the Stable Diffusion 2.1 as the prior. Our FlowDreamer employs the Stable Diffusion 2.1 as the prior. As shown in Figure 6(a), our method generates objects that match well with the input text prompts and exhibit realistic textures. For example, our generated 'pumpkin' is of high fidelity, and only our method generates the 'spiders', which matches the text prompt 'plastic' for the first prompt. The 'origami pig' has rich details, such as its eyes and creases, which are relatively realistic for the second prompt. Although DreamGaussian (Tang et al., 2023) and GaussianDreamer (Yi et al., 2023) require comparatively less time, their results are generally subpar. Our FlowDreamer shows an overall improvement in terms of visual quality and textural details.

**NeRF Generation.** We compare our FlowDreamer with DreamFusion (Poole et al., 2022), Prolific-Dreamer (Wang et al., 2024a), Consistent3D (Wu et al., 2024) in NeRF. Other SoTA baselines (Poole et al., 2022; Wang et al., 2024a; Wu et al., 2024) reimplemented by Three-studio (Guo et al., 2023) codebase and employ Stable Diffusion 2.1 for the prior. Our FlowDreamer employs the Stable Diffusion 2.1 as the prior. As shown in Figure 6(b), our FlowDreamer achieves results with high fidelity and accurate text alignment. For example, the 'saguaro cactus' and 'clay pot' exhibit more detail and greater visual quality for the first prompt. Only FlowDreamer does not render the 'octopus' and 'harp' as a single object for the second prompt. Our FlowDreamer takes only more time than DreamFusion Poole et al. (2022), but the quality of DreamFusion's results is limited. (For more results, please refer to the supplementary material.)

### 4.2 Quantitative Comparisons

We use CLIP (Radford et al., 2021) similarity to quantitatively evaluate our method under either NeRF (Mildenhall et al., 2021) or 3D GS (Kerbl et al., 2023) settings.

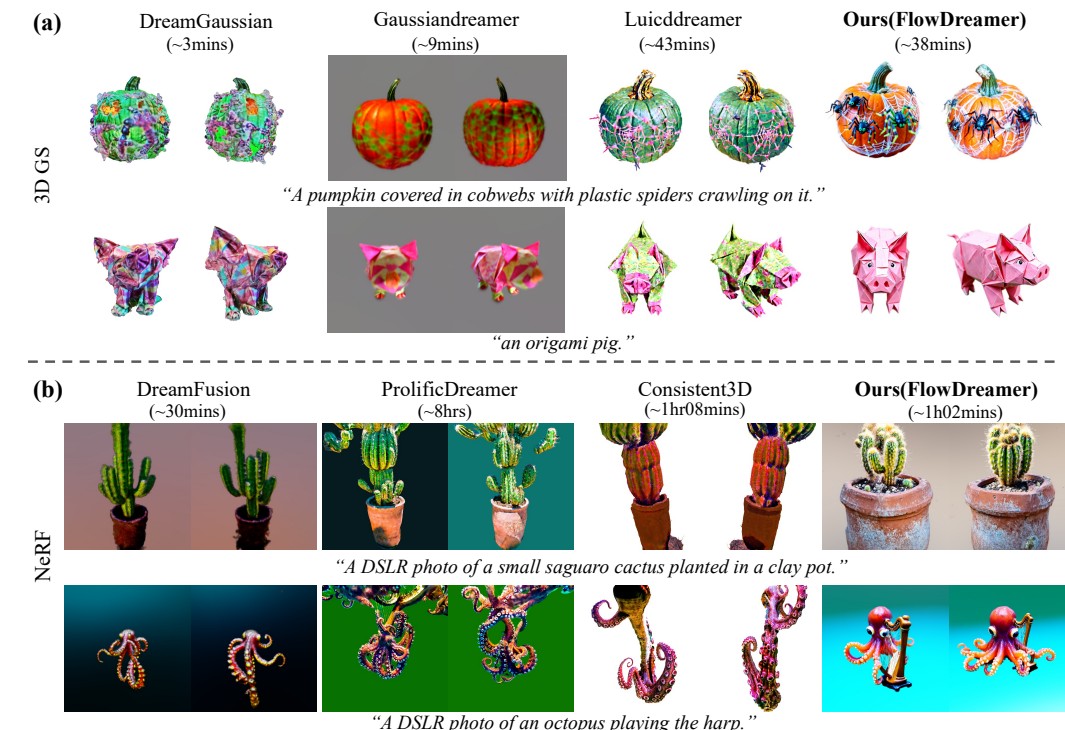

Figure 6: Qualitative comparison under 3D GS and NeRF generation setting. Our FlowDreamer generates objects with finer details.

The results of 3D representations with NeRF come from implementation in (Guo et al., 2023). The results of 3D representation with 3D GS are from their official implementation. The prompts of NeRF results are from Dream-Fusion, and the prompts of 3D GS are from LucidDreamer Liang et al. (2023) and ChatGPT.

Table 1: Quantitative comparisons on CLIP (Radford et al., 2021) similarity with other methods in NeRF generation.

| Methods | ViT-B-32 | ViT-L-14 | ViT-g-14 |
|---|---|---|---|
| Dreamfusion (Poole et al., 2022) | 30.13 | 29.70 | 29.49 |
| Prolificdreamer (Wang et al., 2024a) | 32.62 | 32.55 | 31.60 |
| Consistent3D (Wu et al., 2024) | 32.34 | 32.56 | 32.01 |
| **Ours** | **34.96** | **34.19** | **34.58** |

We randomly choose 26 prompts each to compare in 3D GS and NeRF. We randomly select 12 from the rendered images. The rendered images are from azimuth angles from -180 to 180 degrees with a fixed elevation of 15 degrees for both NeRF and 3D GS. We use three CLIP models from OpenCLIP (Ilharco et al., 2021), ViT-B-32, ViT-L-14, and ViT-g-14, to calculate the CLIP similarity. Our method demonstrates better CLIP similarities both in NeRF and in 3D GS scenarios.

As shown in Tab. 1, our CLIP similarity achieved the best results across all three CLIP models, with the largest margin of 2.57 over the second-best result in ViT-g-14 in NeRF results. And Tab. 2 shows that our CLIP similarity also achieved the best results compared to other methods. In particular, it exceeds the LucidDreamer result by 1.89 in ViT-B-32.

Table 2: Quantitative comparisons on CLIP (Radford et al., 2021) similarity with other methods in 3D Gaussian splatting generation.

| Methods | ViT-B-32 | ViT-L-14 | ViT-g-14 |
|---|---|---|---|
| DreamGaussian (Tang et al., 2023) | 22.94 | 23.50 | 20.76 |
| GaussianDreamer (Yi et al., 2023) | 28.55 | 29.03 | 26.98 |
| LucidDreamer (Liang et al., 2023) | 28.81 | 29.78 | 28.97 |
| **Ours** | **30.70** | **30.49** | **30.66** |

### 4.3 EXPERIMENTAL INSIGHTS OF OUR FLOWDREAMER

**3D generation with rectified flow prior.** To better demonstrate the effectiveness of our method, we replace the SOTA method LucidDreamer's diffusion prior with the rectified flow prior. *The derivation process is provided in the supplementary material.* We refer to the ISM loss of Lucid-Dreamer with vector field as VF-ISM. Figure 7 demonstrates that our method can generate results with finer details and more realistic textures compared with VFDS and VF-ISM in both 3D GS and

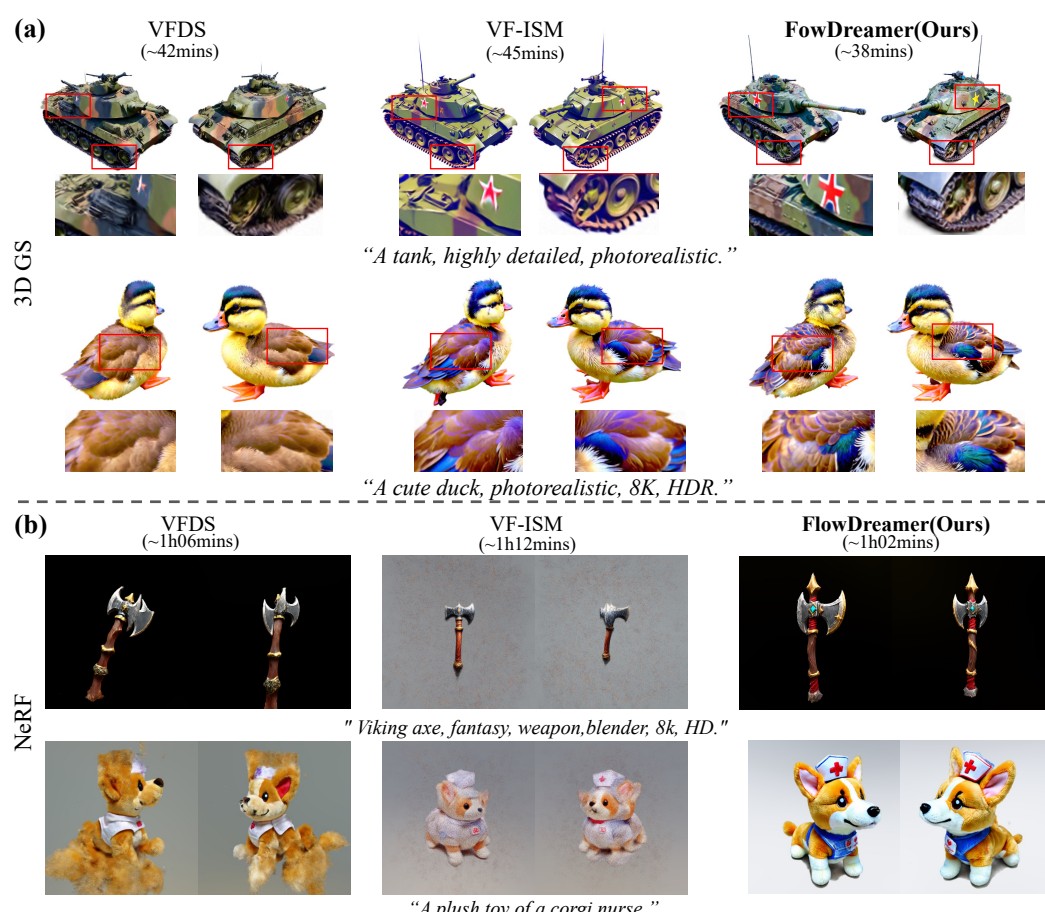

Figure 7: Comparison with other baseline methods using the same rectified flow prior under 3D GS and NeRF generation settings. Our 3D results are with richer details and more realistic colors.

NeRF results. FlowDreamer achieves convergence in 3D GS and NeRF faster than VF-ISM, while demonstrating superior details and more realistic shapes.

For instance, Figure 7(a) in 3D GS indicate that the tank's tracks and the duck's feathers appear heavily blurred in VFDS. Furthermore, the tank's color lacks realism, and the duck's back is somewhat oversaturated in VF-ISM. Our FlowDreamer not only generates the tank and the duck with richer details but also achieves a more realistic overall appearance. In addition, the Figure 7(b) NeRF results reveal that the axe shape and details are improved, whereas the corgi in VFDS exhibits excessive smoothing, and the corgi in VF-ISM presents some noise. The shape and details of our corgi remain relatively satisfactory.

**Impacts of different Classifier-Free Guidance (CFG) scales.** We check the impact of CFG (see Figure 8). The results indicate that we achieve good performance across various CFG scales, demonstrating strong robustness to different CFG scales compared with VFDS.

**Impacts of different Number of Function Evaluations (NFE).** As NFE increases, the training time also increases, and the generated objects exhibit more details and more complex structures (see Figure 9(a)). For example, the structure complexity of the front hood of the LEGO car gradually increases. However, Even with a small NFE, wherein the *push-backward* process has a small cost, our method can still train a 3D model with good performance.

**Impacts of various sampling methods.** We test three sampling methods, namely first-order Eulder, second-order midpoint, and fourth-order Runge-Kutta. Experimental results in Figure 9(b) show that higher-order solvers do not necessarily yield better performance. For example, the total

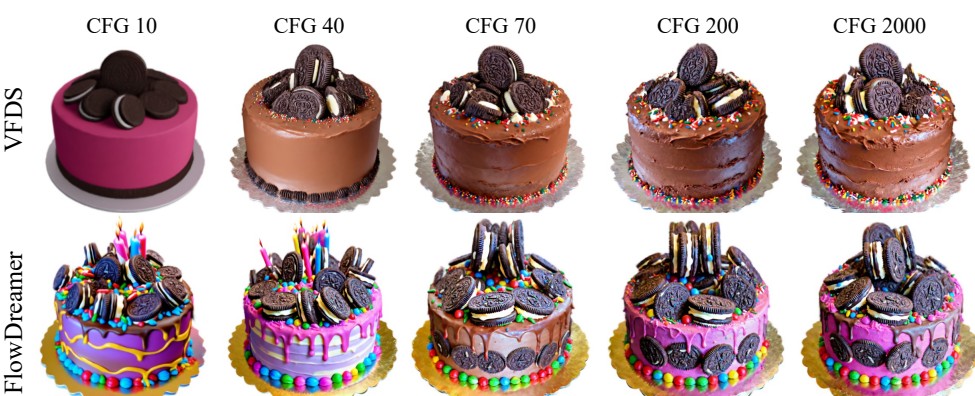

Figure 8: A comparison between our initial framework VFDS (upper) and FlowDreamer (bottom) with different scales of CFG. The results generated by FlowDreamer contain more detailed features. **Prompt**: *"A cake filled with Oreos, highly detailed, photorealistic."*

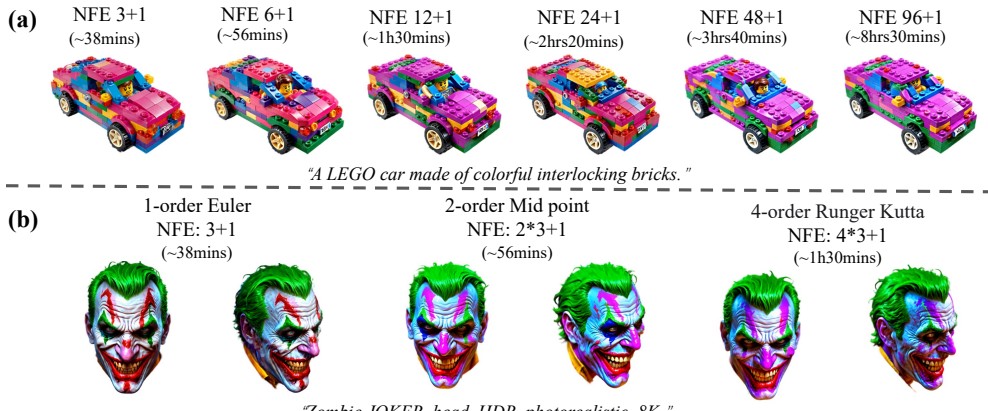

Figure 9: Impacts of different NFEs and sampling methods. "NFE N+1" denotes using $N$ steps of iteration for *push-backward*, and 1 step for gradient calculation. 'NFE: 2*3+1' indicates that the second-order method requires 2 inferences per iteration. The total *push-backward* takes 3 iterations.

*push-backward* process takes three iterations, and the Euler method actually produces more realistic results while requiring the least amount of time. *For more experimental results, please refer to the supplementary material.*

## 5 CONCLUSION

In this paper, we explored a new direction by using the rectified flow model as an alternative prior to text-to-3D generation. We developed a mathematical analysis to adapt SDS to rectified flow model, resulting in the initial VFDS framework. However, VFDS still leads to over-smoothing. We analyzed this issue from the perspective of ODE trajectories and proposed FlowDreamer, a text-to-3D framework with a new UCM loss. Extensive experiments showed that FlowDreamer achieves high-fidelity results with richer details and faster convergence in both NeRF and 3D GS settings. We also highlighted open questions, such as initialization issues for NeRF and noise search sampling. **Limitation.** The Jabus problem still exists; simply adding words like 'front view,' 'back view,' and 'side view' to the prompt is insufficient for supervising the generation view. Although we attempt to mitigate this issue using Perpneg (Armandpour et al., 2023), it still occasionally occurs. We consider solving the Jabus problem thoroughly as a focus for the future work.

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
