# OpenReview forum: "FLOWDREAMER: EXPLORING HIGH FIDELITY TEXT-TO-3D GENERATION VIA RECTIFIED FLOW"
_ICLR.cc/2025/Conference — Submitted to ICLR 2025_

### Official Review · Reviewer_pQyS · 2024-10-19

**Soundness:** 2
**Presentation:** 3
**Contribution:** 1
**Rating:** 5
**Confidence:** 4

**Summary:**

This paper presents Vector Field Distillation Sampling (VFDS) to integrate Score Distillation Sampling (SDS) with rectified flow models. The authors further enhance VFDS by substituting randomly sampled noise with pushed-back noise. The authors then experiment with rectified flow model SDv3 to validate their algorithm.

**Strengths:**

1. The qualititive results are generally good. And the presentation is clear.

2. The authors conducted experiments across various representations, including 3D-GS and NeRF, to validate their algorithm.

**Weaknesses:**

1. **Lack of Novelty**: The paper presents very limited novelty, as rectified flow is merely a specific case within the broader family of diffusion models. Generalizing SDS-like methods to rectified flow is trivial. Specifically, for the forward diffusion process:

   $$
   \boldsymbol{x}_t = a_t \boldsymbol{x}_0 + b_t \boldsymbol{\epsilon}, \tag{1}
   $$

   Rectified Flow sets $ a_t = 1-t $ and $ b_t = t $. With this schedule, the prediction of rectified flow model SDv3 can be reformulated to epsilon prediction according to the equation below Eq. (12) in SDv3 [1]:

   $$
   \boldsymbol{\epsilon}_\phi = (1-t) \boldsymbol{v} _\phi + \boldsymbol{x}_t. \tag{2}
   $$

   This means that all existing SDS-like methods (which are usually formulated in epsilon prediction) can be effectively adapted to the rectified flow model SDv3 by changing the prediction type using Eq. 2. Consequently, re-deriving VFDS and VF-ISM in the paper is unnecessary, as they are trivially equivalent to SDS and ISM. This re-derivation may not offer substantial or particularly inspiring insights. The authors need to clarify what specific novel contributions or challenges they encountered in applying this approach to text-to-3D generation.

2. **Unfair Comparison**: The comparisons presented in this paper appear to be unfair. As stated in Section 4.1, the baselines utilize the SDv2.1 prior, while this work employs the SDv3 diffusion prior as claimed in Appendix section 2. The improvements reflected in the quantitative results in Tables 1 and 2 likely stem from the upgrade of the base diffusion model. Furthermore, as noted in weakness 1, upgrading other baselines to SDv3 is almost trivial. There should be no hindrance to adopting the same base diffusion model for a fair comparison. The authors should include equitable comparisons in their paper.

3. **Limited Novelty of UCM Loss**: Given the triviality of upgrading SDS-like methods to rectified flow models, the proposed UCM loss bears a strong resemblance to the loss used in ISM, further limiting the contribution of this paper. Both methods employ an inversion process to enhance the noising methods. The primary distinction between UCM and prior work ISM appears to be that ISM reverses to timestep $ t $, while UCM inverts to the maximum timestep and combines the noise with the rendered image at timestep $ t $. The authors should further explain on the difference between ISM and their UCM loss and the potiential improvements of UCM compaired with ISM. The result presented in Fig. 7 demonstrated some effectiveness, but further comparison are necessary to enhance the arguments. An additional fair quantitative comparison is necessary to demonstrate the improvements of UCM, and I would likely adjust my rating based on the outcomes of these experiments.

**Questions:**

1. Have I accurately captured the main difference between the proposed algorithm and ISM in weakness 2? I suggest that the authors include a table comparing their methods with ISM for enhanced clarity, given the significant resemblance.

2. Did the authors encounter any challenges when upgrading the base diffusion model to SDv3 for the other baselines using Eq. (2)?

---

> ### Author Response · Authors · 2024-11-20
> **To reviewer pQyS**
>
> Dear reviewer pQyS
>
> ### W1.
> Thank you for your suggestion. Our re-derivation is simply aiming at providing a more rigorous explanation of the final results.
> And we only put it into the appendix.
>
> ### W2.
> Thank you for your suggestion. We have already conducted a fairer comparison. Please refer to the unified response to all reviewers and Fig.17 to Fig.29. of the revised manuscript. Furthermore, the final experiments also demonstrate that our FlowDreamer achieves better results.
>
> ### W3, Q1.
> Thank you for your suggestion very much. We have already conducted a comparison. UCM and ISM differ in different approaches to optimization and different NFE, and whether to transition from images to noise, and so on. Furthermore, the final experiments also demonstrate that our FlowDreamer achieves better results.
>
> ### Q2.
> Thank you for your suggestion. When we replaced other diffusion model-based methods with SDv3, we found that some results were actually worse, possibly because the optimization 3D model parameters were not adjusted specifically for SDv3.
> Once again, thank you for your suggestion, which has been very helpful. Thank you.
>
> I am glad to discuss further with you if you have any other concerns.

---

> > ### Comment · Reviewer_pQyS · 2024-11-22
> > **Response to Submission2782 Authors**
> >
> > I find the samples presented in Fig. 17 to Fig. 29 helpful for evaluating performance differences. While there appear to be some improvements in the results, the observed differences do not seem particularly significant. A quantitative evaluation would provide a clearer and more robust assessment.
> >
> > My primary concern remains the distinction between ISM and UCM loss, as identifying this difference is critical to highlighting the contribution of this work. From the Unified Response, I gather that the main differences are: (1) ISM discards $\eta_t$, and (2) the calculation of $x_t$ differs.
> >
> > Regarding (1), ISM appears to have already performed an ablation study on the impact of $\eta_t$ in their supplementary material, justifying their decision to discard this term. While the context may differ in the Rectified Flow model, an ablation study specific to this setting would provide valuable insights.
> >
> > Regarding (2), the difference in the computation of $x_t$ seems subtle, as the push-backward process could be interpreted as a DDIM inversion to the maximum timestep $𝑡=1$. However, the implications of this computational difference deserve further theoretical discussion or experimental ablation to confirm its significance and contribution to this work.

---

> > > ### Author Response · Authors · 2024-11-23
> > > **To reviewer pQyS**
> > >
> > > Thank you for your suggestions.
> > > We have conducted a quantitative evaluation of the latest results and have included them in the **Unified Response**.
> > >
> > > Regarding (1)
> > >
> > > As suggested by Reviewer 3jWE, we conducted ablation experiments which revealed that ignoring $\eta_t$ actually leads to worse performance, resulting in over-smoothed outcomes(refer to Fig.15). The loss function in ISM is applied over intervals such as $x_s$ and $x_t$, which inherently generates $\eta_t$ terms. These $\eta_t$ terms are all of the same magnitude, and while ISM showed that ignoring them empirically leads to better results, it did not provide a clear theoretical explanation.
> > > In contrast, our UCM loss does not involve intervals and therefore does not generate redundant terms like $\eta_t$, making it more straightforward and intuitive. With this streamlined design, the equation in Eq. 6 of our paper can be implemented using any ODE solver. Moreover, this approach does not require the assumptions made by ISM to ignore $\eta_t$. As a result, our method is more accurate and achieves better performance.
> > >
> > > Regarding (2)
> > >
> > > We performed DDIM inversion up to the maximum timestep $t$, specifically $t=1$. Please refer to Fig.30. The experiments show that our method achieves results even when using DDIM inversion. However, as the number of steps increases, the image noise introduced by DDIM inversion grows, particularly under smaller CFG values. In contrast, our Push-Backward method remains more stable.
> > > We attribute this stability to the design of Rectified Flow, which inherently considers reversibility. Since it employs linear interpolation to obtain $x_t$, its formulation is symmetric, which is beneficial for maintaining reversibility. Consequently, Push-Backward demonstrates greater stability in these scenarios, **further proving the significance of our UCM loss design under the Rectified Flow framework.**

---

> > > > ### Comment · Reviewer_pQyS · 2024-11-25
> > > >
> > > > I appreciate the authors' detailed responses, which have clarified several points and addressed some of my initial concerns. However, after carefully considering the replies, I am inclined to maintain my current score rather than increasing it. That said, I will not block acceptance if the other reviewers express strongly positive opinions. While I generally find the experimental outcomes of this paper better, the improvements appear not significant to me.
> > > >
> > > > My primary concern remains with the theoretical aspects of the paper (as also noted by Reviewer W25F). Specifically, I believe the current version lacks a thorough comparison with prior work, particularly ISM. The re-derivation of the UCM loss in Sections 3.1 and 3.2 bears significant similarities to the loss derivation in ISM but results in a slightly different formulation. Furthermore, the differences highlighted by the authors do not demonstrate any clear theoretical advantages of their methods. Without a sufficiently detailed theoretical analysis or interpretation of these differences, this aspect of the paper feels underdeveloped and leaves a critical gap in the narrative.
> > > >
> > > > Additionally, as Reviewer CLaj pointed out, FM models are a reparameterization of diffusion models. While the trajectory may indeed be straighter, the paper does not clearly articulate the advantages of adapting the proposed methods to FM models. The authors note that their methods are also compatible with standard diffusion models like SDv2. If this reparameterization does not confer distinct benefits, presenting the work in the context of general diffusion models might provide better clarity and broader applicability. Alternatively, if the FM parameterization offers specific advantages, the authors should clearly explain how their methods benefit from this reparameterization.
> > > >
> > > > I still have some questions regarding the paper (these questions do not influence my score):
> > > >
> > > > 1. **Figure 30 experiment setup**: The experimental setup for Figure 30 is somewhat unclear. A more detailed explanation would enhance understanding. Specifically, did the authors fix the time step of $ x_t $ to $ t = 1 $ for DDIM inversion while using an annealing or randomized schedule for the push-backward process?
> > > >
> > > > 2. **Definition of push-backward in general diffusion models**: What exactly does "push-backward" mean in the context of general diffusion model parameterizations? Does it refer to DDIM inversion to the maximum timestep $ T $ (to pure noise $b_T \boldsymbol{\epsilon}$), followed by a linear combination of the noise and variable as $ \boldsymbol{x}_t = a_t \boldsymbol{x}_0 + b_t \boldsymbol{\epsilon} $ at timestep $ t $? I suspect I may have misunderstood the definition, so I would appreciate any corrections or clarifications.

---

> > > > > ### Author Response · Authors · 2024-11-25
> > > > > **To reviewer pQyS**
> > > > >
> > > > > Q: Furthermore, the differences highlighted by the authors do not demonstrate any clear theoretical advantages of their methods. Without a sufficiently detailed theoretical analysis or interpretation of these differences, this aspect of the paper feels underdeveloped and leaves a critical gap in the narrative.
> > > > >
> > > > > We conducted ablation experiments which revealed that ignoring $\eta_t$ actually leads to worse performance, resulting in over-smoothed outcomes(refer to Fig.15). The loss function in ISM is applied over intervals such as $x_s$ and $x_t$, which inherently generates $\eta_t$ terms. These $\eta_t$ terms are all of the same magnitude, and while ISM showed that ignoring them empirically leads to better results, it did not provide a clear theoretical explanation.
> > > > > In contrast, our UCM loss does not involve intervals and therefore does not generate redundant terms like $\eta_t$, making it more straightforward and intuitive. With this streamlined design, the equation in Eq. 6 of our paper can be implemented using any ODE solver. Moreover, this approach does not require the assumptions made by ISM to ignore $\eta_t$. As a result, our method is more accurate and achieves better performance.
> > > > >
> > > > > Q: Alternatively, if the FM parameterization offers specific advantages, the authors should clearly explain how their methods benefit from this reparameterization.
> > > > >
> > > > > Below, we provide a theoretical analysis. In SDS, randomly sampled Gaussian noise and randomly sampling of $t$, and the gradient of $\epsilon_\theta(x_t) - \epsilon$ is computed to obtain the score direction, which is then used to update the parameters of the 3D model. We need to gradually improve the quality of the rendered images. The score direction represents the gradient direction of $x_t$ and is used to update the rendered image $x$. In curved diffusion trajectories, the gradient direction of $x_t$ and the gradient direction of $x$ are often inconsistent. In contrast, RF, due to its straight rectified flow trajectory, ensures that these directions are consistent, leading to better updates.
> > > > >
> > > > > Additionally, in RF, $t \epsilon + (1-t)x = x_t$, the linear interpolation inherently makes the image $x$ and $\epsilon$ symmetric. Therefore, using UCM to find the noise becomes very straightforward and reasonable.
> > > > >
> > > > > Q:Figure 30 experiment setup
> > > > >
> > > > > DDIM inversion and push-backward both aim to reach \(t=1\).
> > > > >
> > > > > Q:Definition of push-backward in general diffusion models:
> > > > > "push-backward" means that the process from image $x$ to search coupled noise $\epsilon$.
> > > > > UCM loss use coupled noise to replace randomly sampled noise in VFDS.
> > > > >
> > > > > Push-backward only requires taking an Euler step at each iteration to find the corresponding noise. This process is entirely continuous, with the distance and sampling method entirely determined by your settings.  However, DDIM is discrete. DDIM inversion requires first finding $x_0$at each step and then using the diffusion forward process to obtain the next noisy latent $x_{t-1}$.
> > > > >
> > > > > We are glad to discuss further with you if you have any other concerns.

---

> > > > > > ### Comment · Reviewer_pQyS · 2024-11-26
> > > > > >
> > > > > > I have decided to maintain my current score, but I remain open to further discussion.
> > > > > >
> > > > > > I still feel the authors have not provided a clear justification for the benefits of using RF/FM models. I agree with Reviewers bjYV and CLaj that RF models do not guarantee a straight-line trajectory (even though the trajectory may be straighter). Moreover, the reverse property is not unique to RF models, as both RF and other diffusion models require a multi-step discretized inversion process.
> > > > > >
> > > > > > > *"In curved diffusion trajectories, the gradient direction of $x_t$ and the gradient direction of $x$ are often inconsistent. In contrast, RF, due to its straight rectified flow trajectory, ensures that these directions are consistent, leading to better updates."*
> > > > > >
> > > > > > This explanation does not clearly justify how a straighter trajectory ensures consistent gradient directions. I believe the gradient directions in the RF model can still be inconsistent since its trajectory is not perfectly straight.
> > > > > >
> > > > > > > *"Push-backward only requires taking an Euler step at each iteration to find the corresponding noise. This process is entirely continuous, with the distance and sampling method entirely determined by your settings."*
> > > > > >
> > > > > > How can the push-backward process be achieved using just one Euler step? I note that Equation 6 in the paper describes a multi-step process, and the NFE is reported as (3+1) rather than (1+1). Could you clarify this discrepancy?

---

> > > > > > > ### Author Response · Authors · 2024-11-26
> > > > > > > **To reviewer pQyS.**
> > > > > > >
> > > > > > > As you mentioned, RF is not strictly a straight line, so whenever we analyze the straight line in our paper, we always include the phrase "under ideal circumstances." The reverse property is not unique to RF models; we just wanted to explore text-to-3D from the RF perspective. As you said, any Diffusion model can be parameterized and converted into an RF, Moreover, they can all be sampled using the ODE approach. But we believe that exploring some unknown areas within the RF framework is also crucial.
> > > > > > >
> > > > > > > Q:NFE is reported as (3+1) rather than (1+1).
> > > > > > >
> > > > > > > As we mentioned, we all agree that RF is not a strictly straight line, which is why we used 3 steps. However, in theory, 3 steps are not sufficient to find a coupled noise. Our experiments show that 3 steps can still achieve a relatively good result, and we respect the experimental outcomes in presenting our results.
> > > > > > >
> > > > > > > **We believe our goal in this discussion is to enhance both parties' understanding of the topic. Therefore, We are glad to discuss further with you if you have any other concerns.**

---

### Official Review · Reviewer_CLaj · 2024-10-27

**Soundness:** 2
**Presentation:** 2
**Contribution:** 2
**Rating:** 3
**Confidence:** 4

**Summary:**

This paper introduces a new framework called FlowDreamer, using a FM based model to update a 3D generator, e.g., Nerf or GS.

The whole framework mainly contains 2 part:
1. a score distillation based on FM model (named VFDS in paper) and
2. a push-back method (named UCM in paper) to align noise and rendered scene.


Finally, extensive experiments demonstrate that the FlowDreamer framework achieves high-fidelity and fast convergence across various 3D generation settings, such as NeRF and 3D GS.

**Strengths:**

This paper involves extensive engineering work to distill a high-quality 3D generator using the SD3 model as the foundational framework.

**Weaknesses:**

I have significant concerns regarding various aspects of this paper.

1. The paper suffers from a lack of citations, particularly in relation to the references concerning UCM. Similar methods have already been extensively applied in DMD2 [1] and Imagen Flash [2]. The author state UCM is improved from ISM but I rather suggest check the given reference and rethink the novelty of UCM.

2. The authors have not provided adequate experimental evidence to validate the effectiveness of the individual components of their approach. Although they demonstrate strong overall performance metrics, there is no experimental support showing that the use of **straight trajectories contributes to performance improvements**.

    As FM is simply a parameterized special case of Diffusion. I am not convinced that straight trajectories will have a substantial effect on score distillation. Moreover, in the case of Diffusion, higher-order scores also follow straight trajectories. As such, I would require the authors to present experimental evidence demonstrating the improvements claimed to result from the use of straight trajectories.

3. Furthermore, we observed that the methods used for comparison were based on SD2.1, while the authors employed SD3 as the teacher model, which constitutes an inherently unfair comparison. Notably, this issue was not mentioned in the main manuscript. I am concerned that the observed strong alignment between the text and 3D objects may be attributed to SD3’s MMDIT, rather than the method introduced by the authors.




[1] Yin, Tianwei, et al. "Improved Distribution Matching Distillation for Fast Image Synthesis." arXiv preprint arXiv:2405.14867 (2024).

[2] Kohler, Jonas, et al. "Imagine flash: Accelerating emu diffusion models with backward distillation." arXiv preprint arXiv:2405.05224 (2024).

**Questions:**

see weakness.

I believe that this work demonstrates the potential of distilling a 3D generator using SD3 as the teacher model. However, the current claims put forward in the paper lack strong experimental validation, and I hold reservations about some of the arguments presented by the authors. For these reasons, I recommend rejecting this paper.

---

> ### Author Response · Authors · 2024-11-20
> **To reviewer CLaj**
>
> Dear reviewer CLaj
>
> ### W1.
> Thank you for suggesting these two interesting works.
> However, Our FlowDreamer focus on text-to-3D and are more concerned with identifying the cause of over-smoothing. We found that the smoothing effect in the rectified flow is caused by randomly sampling Gaussian noise and directly addressed this issue by replacing the noise. We will also include these papers in the related work section later and provide a discussion on them in the next few days.
>
> ### W2.
> Thank you for your suggestion.
> We have also provided experiments to demonstrate that straight trajectories can accelerate convergence. We conducted experiments using Diffusion1.5 and the 2-rectified-flow from Instaflow, as 2-rectified-flow distilled a straighter trajectory from Diffusion1.5 while maintaining the same model architecture. In the experiments, we observed that 2-rectified-flow converged faster. However, your insights regarding the understanding of straight trajectories prompted us to reflect further. We will express this more rigorously in future versions within a few days.
>
> ### W3.
> Thank you for your suggestion. To ensure a fairer comparison, we have conducted a large number of additional experiments. Please refer to the unified response to all reviewers. Furthermore, the final experiments also demonstrate that our FlowDreamer achieves better results.
>
> I am glad to discuss further with you if you have any other concerns.

---

> > ### Comment · Reviewer_CLaj · 2024-11-23
> >
> > Thanks the reply from authors.
> >
> > For W2, RF2 is a distilled version based on SD1.5, which demands substantially more computational resources compared to SD1.5. This setup raises challenges in validating the paper’s hypothesis that a linear trajectory can accelerate convergence. I would encourage the authors to consider conducting fair comparative experiments or providing theoretical analysis to support this claim. Specifically, if the RF’s DDIM pairwise distillation scheme were employed without modifying the model parameterization during distillation, would it still achieve comparable results?
> >
> > For W3, I still could not find any results demonstrating how the proposed method performs on SD1.5 or 2.1. If such results are unavailable, could the authors clarify why this approach cannot be applied to SD1.5 or 2.1?
> >
> > It is noteworthy that the authors have incorporated results for consistency3D. It might be beneficial for the authors to reference [1] and explore state-of-the-art text-to-3D approaches related to consistency.
> >
> > [1] Li, Zongrui, et al. "Connecting Consistency Distillation to Score Distillation for Text-to-3D Generation." European Conference on Computer Vision. Springer, Cham, 2025.

---

> ### Author Response · Authors · 2024-11-24
> **To reviewer CLaj**
>
> Thank you for your suggestion.
>
> For W2.
>
> As you mentioned, the trajectory of 1-rectified-flow (without distillation) is not very straight, so in our paper, we included the phrase "under ideal circumstances" in our analysis. In the comparison between SDS and VFDS, [2]Flow Matching also pointed out that flow trajectories are straight, whereas diffusion paths result in curved paths. As mentioned in [1], 1-rectified-flow still contains a proportion of straight trajectory, which are straighter than those in diffusion. Therefore, we stated that VFDS's optimization direction is more consistent. Of course, your suggestion is very insightful, and we will express this more rigorously in the paper. From [3]Instaflow, we use 2-rectified-flow (a method that better meets our assumptions of ideal circumstances) for our experiments. The goal was to achieve straighter rectified flow, which currently most methods address through distillation. Distillation requires more computational resources, and it is unavoidable. We appreciate your suggestion and will express this more rigorously in future papers.
>
> For W3
>
> We focus on exploring text-to-3D generation from the perspective of rectified flow. Therefore, we did not provide many experiments on SD2.1 or SD1.5. However, Following the suggestion of reviewer pQyS, we also demonstrated that our method achieves certain ablation studies using SD2.1(Please ref to Fig.30). Thank you for your suggestion.
>
> The paper you mentioned is indeed very interesting and is also an ECCV paper and a concurrent work with ours. However, our focus is on exploring text-to-3D generation via rectified flow and analyzing the fundamental cause of VFDS's over-smoothing from the perspective of ODE trajectories. We then addressed this issue using the properties of rectified flow. We will discuss and compare this paper in the related work section in future versions.
>
> **We are glad to discuss further with you if you have any other concerns.**
>
> [1]Liu, Xingchao, Chengyue Gong, and Qiang Liu. "Flow straight and fast: Learning to generate and transfer data with rectified flow." arXiv preprint arXiv:2209.03003 (2022).
>
> [2]Lipman, Yaron, et al. "Flow matching for generative modeling." arXiv preprint arXiv:2210.02747 (2022).
>
> [3]Liu, Xingchao, et al. "Instaflow: One step is enough for high-quality diffusion-based text-to-image generation." The Twelfth International Conference on Learning Representations. 2023.

---

> ### Comment · Reviewer_CLaj · 2024-11-25
>
> Thanks for your reply.
> However I have to say my concerns are still there:
>
> 1. Why is this approach inapplicable to SD1.5 or 2.1? Particularly when RF is a specific parameterization version of diffusion.
>
> 2. What experimental designs or theoretical justifications, if any, could more convincingly demonstrate that the so-called linear trajectory improves the distillation efficiency of methods like SDS?

---

> ### Author Response · Authors · 2024-11-25
> **To reviewer CLaj**
>
> Q1:Why is this approach inapplicable to SD1.5 or 2.1? Particularly when RF is a specific parameterization version of diffusion.
>
> As our paper title, we focus on exploring text-to-3D generation from the perspective of rectified flow. However, we also demonstrated that our method achieves certain ablation studies using SD2.1(Please ref to Fig.30). We use RF to search the coupled noise, it's more stable.
>
> Q2:What experimental designs or theoretical justifications, if any, could more convincingly demonstrate that the so-called linear trajectory improves the distillation efficiency of methods like SDS?
>
> We have already implemented 2-rectified-flow, which represents a straighter trajectory. With this straighter trajectory as the model prior, experiments have demonstrated that leads to significantly faster convergence(Please ref to Fig.14). Furthermore, in our original paper, Fig. 4 provides an analysis explaining why RF enables faster convergence.
>
> Below, we provide a theoretical analysis. In SDS, randomly sampled Gaussian noise and randomly sampling of $t$, and the gradient of $\epsilon_\theta(x_t) - \epsilon$ is computed to obtain the score direction, which is then used to update the parameters of the 3D model. We need to gradually improve the quality of the rendered images. The score direction represents the gradient direction of $x_t$ and is used to update the rendered image $x$. In curved diffusion trajectories, the gradient direction of $x_t$ and the gradient direction of $x$ are often inconsistent. In contrast, RF, due to its straight rectified flow trajectory, ensures that these directions are consistent, leading to better updates.
>
> We are glad to discuss further with you if you have any other concerns.

---

### Official Review · Reviewer_W25F · 2024-11-04

**Soundness:** 2
**Presentation:** 3
**Contribution:** 2
**Rating:** 5
**Confidence:** 5

**Summary:**

The paper proposes using a flow matching model as a replacement for the diffusion model within the SDS framework. It offers an analysis of data and noise coupling alongside the reversible properties in flow matching, aiming to mitigate the noisy gradient signals inherent in the SDS framework by back-tracking the corresponding noise for each current rendered view. Empirical results demonstrate an improved CLIP score compared to previous approaches.

**Strengths:**

- the paper is written in good clarity
- one of the first papers incoporating flow matching to SDS loss and exploit the flow matching coupling/reversible property to improve SDS stability

**Weaknesses:**

- The paper asserts that “the trajectory of the rectified flow model is straight, v_phi(xt, t), and remains approximately constant for different t under ideal conditions.” However, this assumption appears incorrect. Even if trained with a straight velocity, the rectified flow typically displays a curved trajectory, with flow paths re-routed at intersections to prevent crossings. This phenomenon is discussed and illustrated in Fig. 2(b) [1].
- Given this questionable assumption, it also raises doubts about the claimed advantages of VFDS in speed and stability over SDS. Although the paper suggests that VFDS converges more rapidly than the diffusion model and that FlowDreamer converges faster than VFDS, it provides only one qualitative example in the appendix without sufficient large-scale quantitative evidence or convincing benchmark metrics.
- The authors state, “in contrast, we propose a novel UCM loss based on the push-backward process to identify corresponding noise, rather than using randomly sampled noise as in rectified flow-based models.” However, prior work in LucidDreamer demonstrates a similar approach by eliminating randomness in xt through deterministic DDIM inversion, thereby increasing pseudo-GT consistency. This paper should acknowledge the correlation with LucidDreamer and correct its claims. Furthermore, as DDIM is essentially an ODE solver [2], deterministic DDIM inversion closely resembles the noise back-tracking proposed in this work. Both aim to reduce randomness, indicating that the novelty of this approach may be limited.
- Although FlowDreamer is proposed primarily to address over-smoothing issues, the paper provides only three qualitative results in Figures 7 and 8. One could even argue that in Figure 8 the level of detail between VFDS and FlowDreamer appears similar with only different styling, raising questions about the model’s effectiveness in resolving over-smoothing.
- Referring to [3], flow matching models generally perform better than diffusion models. Given the marginal CLIP score improvement in Tables 1 and 2, it remains unclear if the performance gain is due to replacing the diffusion model with the flow matching model.
- The authors highlight a limitation of FlowDreamer in its early stages when the rendered view x falls outside the target data distribution, causing instability in the back-tracked noise epsilon. To address this, a warm-up phase is required, with the initial rendered views needing to be within reasonable limits. However, the paper lacks a detailed ablation study of this two-phase training framework to validate its robustness. For instance, how does varying the warm-up duration impact performance? Is the warm-up phase length dependent on the example’s complexity (e.g., more complex scenes requiring a longer warm-up)?
- ProlificDreamer [4] demonstrates that by using a learnable noise prediction network, the CFG weight can be reduced to a standard range (e.g., 7.5) to yield reasonable results. Given that FlowDreamer is expected to produce more stable noise, it is unclear why a very high CFG value is still required to achieve detailed textures (i.e., ≥40 as shown in Figure 8).

[1] Liu, X., Gong, C. and Liu, Q., 2022. Flow straight and fast: Learning to generate and transfer data with rectified flow. arXiv preprint arXiv:2209.03003.

[2] Lu, C., Zhou, Y., Bao, F., Chen, J., Li, C. and Zhu, J., 2022. Dpm-solver: A fast ode solver for diffusion probabilistic model sampling in around 10 steps. Advances in Neural Information Processing Systems, 35, pp.5775-5787.

[3] Ma, N., Goldstein, M., Albergo, M.S., Boffi, N.M., Vanden-Eijnden, E. and Xie, S., 2024. Sit: Exploring flow and diffusion-based generative models with scalable interpolant transformers. arXiv preprint arXiv:2401.08740.

[4] Wang, Z., Lu, C., Wang, Y., Bao, F., Li, C., Su, H. and Zhu, J., 2024. Prolificdreamer: High-fidelity and diverse text-to-3d generation with variational score distillation. Advances in Neural Information Processing Systems, 36.

**Questions:**

- Can you elaborate on why you assume the trajectory of the rectified flow model remains straight across different t values? How do you address evidence suggesting that rectified flows are typically curved and adjusted to avoid intersection points?
- Could you provide additional quantitative evidence or large-scale experiments to substantiate the claim that VFDS converges faster and is more stable than SDS?
- How does FlowDreamer differenciates with LucidDreamer's deterministic DDIM inversion?
- Could you include a more comprehensive analysis in either 2D/3D to demonstrate FlowDreamer does improve over-smoothing issue significantly compared to VFDS baseline?
- Given that flow matching models often outperform diffusion models [3], how do you confirm that the improvement in CLIP scores (Tables 1 and 2) is due to your proposed methods rather than merely switching to a flow matching model?
- Could you provide an ablation study or detailed analysis to clarify the role of the warm-up phase in the two-stage training framework? Specifically, how does varying the warm-up duration impact model performance and stability?
- Given that FlowDreamer also stabilizes noise, why is a much higher CFG weight (e.g., ≥40) required to achieve detailed textures in FlowDreamer?

---

> ### Author Response · Authors · 2024-11-20
> **To reviewer W25F**
>
> Dear reviewer W25F
>
> ### Q1.
> This is an interesting question. At the time of our analysis, we approached it from a theoretical perspective, but there are some differences between theory and practice. We will provide a more rigorous explanation of this issue in the next few days.
>
> ### Q2.
> Thank you for your suggestion.
> We provided more comparisons on speed among SDS, VFDS, and FlowDreamer. Please refer to Fig.12 and Fig.13. The experimental results clearly indicate that our FlowDreamer converges faster than VFDS, and VFDS converges faster than SDS.
>
> ### Q3.
> Thank you for pointing this out.
> We conducted a more in-depth comparison with ISM. Please refer to the unified response to all reviewers. FlowDreamer and LucidDreamer differ in their perspectives on addressing the problem, approaches to optimization, and other aspects.
>
> ### Q4, Q5.
> Thank you for your suggestion.
> We have already conducted fairer experiments and provided many cases. Please refer to the unified response to all reviewers and Fig.17 to Fig.29. of the revised manuscript. Furthermore, the final experiments also demonstrate that our FlowDreamer achieves better results.
>
> ### Q6.
> Thank you for your suggestion.
> Please refer to Fig.16. As can be seen, without warm-up, the colors appear somewhat saturated. As the number of warm-up steps increases, the results improve, and at 1200 steps, the performance becomes highly stable.
>
> ### Q7.
> Thank you for your suggestion.
> Although FlowDreamer can identify a coupled noise, in our experiments, we only used 3 steps to sample and obtain the coupled noise.  As a result, the coupled noise found is not entirely accurate. For some complex objects, we can enhance their details more effectively by using a larger CFG.
>
>
> If you would like to see more comparison images or have other experimental requests, We are glad to discuss further with you.

---

> ### Comment · Reviewer_W25F · 2024-11-26
>
> I appreciate author's efforts in addressing my original feedbacks. Here are my updated comments:
>
> > This is an interesting question. At the time of our analysis, we approached it from a theoretical perspective, but there are some differences between theory and practice. We will provide a more rigorous explanation of this issue in the next few days.
>
> I did not receive a concrete explanation on this matter.
>
> > We provided more comparisons on speed among SDS, VFDS, and FlowDreamer. Please refer to Fig.12 and Fig.13. The experimental results clearly indicate that our FlowDreamer converges faster than VFDS, and VFDS converges faster than SDS.
>
> I appreciate the authors for providing additional qualitative results. However, the four visual examples provided are insufficient to serve as large-scale quantitative evidence for assessing convergence speed. For instance, tracking a metric like the CLIP score over training iterations on a diverse set of prompts would better reflect the trends in convergence.
>
> > Thank you for pointing this out. We conducted a more in-depth comparison with ISM. Please refer to the unified response to all reviewers. FlowDreamer and LucidDreamer differ in their perspectives on addressing the problem, approaches to optimization, and other aspects.
>
> After reviewing the authors' general comments and discussions with other reviewers, I remain unconvinced regarding the novelty of the proposed UCM loss in comparison to ISM. This is especially true given the observations raised by Reviewer bjYV. As stated in the paper, the reversible property is defined such that ϵ sampled from the Gaussian noise distribution can map to x, while x sampled from the data distribution can also map back to ϵ. Under this definition, both flow matching and DDIM inherently possess this property (as noted by Reviewer bjYV). In essence, LucidDreamer already demonstrated that reducing randomness in xt through deterministic DDIM inversion enhances pseudo-ground-truth consistency, which is closely related to the reversible property mentioned in this paper.
>
> > Thank you for your suggestion. Please refer to Fig.16. As can be seen, without warm-up, the colors appear somewhat saturated. As the number of warm-up steps increases, the results improve, and at 1200 steps, the performance becomes highly stable.
>
> My earlier concerns remain. With only three visual examples and a single-view perspective, it is challenging to draw robust conclusions about the effectiveness or robustness of the proposed two-phase training framework.
>
> > Thank you for your suggestion. Although FlowDreamer can identify a coupled noise, in our experiments, we only used 3 steps to sample and obtain the coupled noise. As a result, the coupled noise found is not entirely accurate. For some complex objects, we can enhance their details more effectively by using a larger CFG.
>
> If the need for a large CFG is primarily due to the choice of using only three steps for noise sampling, does this imply that one could achieve comparable results with a smaller CFG (e.g., 7.5) by increasing the number of noise sampling steps?

---

> > ### Author Response · Authors · 2024-11-26
> > **To reviewer W25F**
> >
> > Thank you for your suggestion.
> >
> > Q:I did not receive a concrete explanation on this matter.
> >
> > We share the same understanding with you and agree that RF is not strictly a straight line. Therefore, in our paper, when analyzing from the perspective of a straight line, we often included the phrase "under ideal circumstances." Instaflow also distilled a relatively straight flow model through distillation, and this flow model can generate an image in just one step. We also provided comparative experiments between SDS and VFDS (please refer to Fig. 14).
> >
> > Q:I appreciate the authors for providing additional qualitative results. However, the four visual examples provided are insufficient to serve as large-scale quantitative evidence for assessing convergence speed. For instance, tracking a metric like the CLIP score over training iterations on a diverse set of prompts would better reflect the trends in convergence.
> >
> > We actually tried using the CLIP score as a threshold and different training iterations to quantify the convergence speed. However, we found that the CLIP score is not very effective at evaluating subtle changes in the image. As you know, the loss curve in this framework also cannot accurately reflect the convergence speed. We have indeed attempted various methods to quantify it, but this specific framework makes quantification particularly challenging. If you would like to see more examples, we can also provide them. However, the current methods are difficult to quantify. We appreciate your understanding.
> >
> > Q:In essence, LucidDreamer already demonstrated that reducing randomness in xt through deterministic DDIM inversion enhances pseudo-ground-truth consistency, which is closely related to the reversible property mentioned in this paper.
> >
> > We conducted ablation experiments which revealed that ignoring $\eta_t$ actually leads to worse performance, resulting in over-smoothed outcomes(refer to Fig.15). The loss function in ISM is applied over intervals such as $x_s$ and $x_t$, which inherently generates $\eta_t$ terms. These $\eta_t$ terms are all of the same magnitude, and while ISM showed that ignoring them empirically leads to better results, it did not provide a clear theoretical explanation.
> >
> > In contrast, our UCM loss does not involve intervals and therefore does not generate redundant terms like $\eta_t$, making it more straightforward and intuitive. With this streamlined design, the equation in Eq. 6 of our paper can be implemented using any ODE solver. Moreover, this approach does not require the assumptions made by ISM to ignore $\eta_t$. As a result, our method is more accurate and achieves better performance.
> >
> >
> > Q:My earlier concerns remain. With only three visual examples and a single-view perspective, it is challenging to draw robust conclusions about the effectiveness or robustness of the proposed two-phase training framework.
> >
> > We admit that our consideration here was insufficient. We view this as a comparison figure for ablation experiments. Thank you for pointing this out. In future experiments, we will ensure that all prompts are completed and provide either a CLIP score or extensive comparison figures to present the results.
> >
> > Q:If the need for a large CFG is primarily due to the choice of using only three steps for noise sampling, does this imply that one could achieve comparable results with a smaller CFG (e.g., 7.5) by increasing the number of noise sampling steps?
> >
> > We have conducted the corresponding ablation experiments, but as the number of steps increases, smaller CFG values show some improvement in results. However, we must acknowledge that the performance boost is not as significant as with three steps at CFG 40.0.
> >
> > **We believe our goal in this discussion is to enhance both parties' understanding of the topic. Therefore, We are glad to discuss further with you if you have any other concerns.**

---

### Official Review · Reviewer_3jWE · 2024-11-04

**Soundness:** 3
**Presentation:** 3
**Contribution:** 3
**Rating:** 6
**Confidence:** 4

**Summary:**

This paper proposed to adopt the pre-trained rectified flow model for text-to-3D generation. Considering the different network predictions, it modified the SDS to match the formulation of rectified flow model . Moreover, push-backward process is used to estimate noise, which provides more consistent supervision signals, leading to highly detailed objects generation.  Comprehensive experiments are conducted to valid the efficacy of proposed methods.

**Strengths:**

1.  This paper claims the first exploration using rectified flow model for text-to-3D generation.
2.  Their results outperform previous SOTA method (i.e., lucidDreamer)
3.  The push-backward process enhances the details of generated objects.

**Weaknesses:**

1. This paper is based on previous observations. For example, the over-smoothed results are caused by noisy gradient. LucidDreamer solved this by replacing sampled random noise with noisy latents derived from DDIM inverse. This paper also use  similar inverse process, i.e., push-backward process, suitable for rectified flow model. Therefore, this article mainly studies how to adapt previous findings to more advanced rectified flow model, which is not interesting enough.
2.  There are several questions about results. 1) In fig.7, does VF-ISM use the same 3D parameters as flowdreamer? 2) LucidDreamer discards some terms for saving time, which is not optimal. Can you show results using Eq.12 in LucidDreamer with the proposed rectified flow model + push-backward process?  3) In Fig. 8, VFDS with large CFG has good details, does that mean UCM loss is not essential? Please show more cases and cases with larger CFG.

**Questions:**

See weakness

---

> ### Author Response · Authors · 2024-11-20
> **To reviewer 3jWE**
>
> Dear reviewer 3jWE
>
> ### W1.
> Thank you for pointing this out.
> We conducted a more in-depth comparison with ISM. Please refer to the unified response to all reviewers. Furthermore, the final experiments also demonstrate that our FlowDreamer achieves better results.
>
> ### W2.
> 1) Yes, the parameters are exactly the same.
> 2) Thank you for your suggestion. We implemented it and found that its performance was actually worse, resulting in even smoother outcomes. Please refer to Fig.15.
> 3) Thank you for your suggestion.
> This time, we provide numerous cases to demonstrate the essentiality of UCM loss. Please refer to Fig.17 to Fig.29. In a more fair experiment, the results still demonstrate that our FlowDreamer performs better.essentiality of UCM loss. Please refer to Fig.17 to Fig.29.
>
> If you would like to see more comparison images or have other experimental requests, We are glad to discuss further with you.

---

> ### Comment · Reviewer_3jWE · 2024-11-26
>
> Thanks for the authors’ clarification and I will keep my score.

---

> > ### Author Response · Authors · 2024-11-26
> > **To Reviewer 3jWE**
> >
> > **We believe our goal in this discussion is to enhance both parties' understanding of the topic. Therefore, We are glad to discuss further with you if you have any other concerns.**

---

### Official Review · Reviewer_bjYV · 2024-11-10

**Soundness:** 2
**Presentation:** 2
**Contribution:** 2
**Rating:** 5
**Confidence:** 4

**Summary:**

This study proposes FlowDreamer to leverage a pretrained text-to-image (T2I) models trained via the rectified flow framework for score distillation sampling. After explaining the training objective of SDS with a flow model, called VFDS, this paper claim that the cause of underperformance results from using multiple noises during optimization. Thus, considering the mapping between a noise and an image, FlowDreamer first use a flow model to conduct the rendered image inversion to noises, and then apply VFDS. FlowDreamer shows better results than existing methods and VFDS for text-to-3D generation

**Strengths:**

S1. This study successfully incorporate a T2I flow model in text-to-3D generation.

S2. Compared to VFDS, the proposed noise sampling techniques shows improvements in generation quality even with a few inversion steps.

**Weaknesses:**

W1. The comparison with existing methods are unfair. Since the existing methods use Stable-Diffusion (SD) 2.1, the performance difference can come from the differences in used T2I models, considering SD 3, which is used for the proposed method, has better T2I performance than SD 2.1.

W2. Unclear contributions. The major contribution lies in the consideration of the characteristics of flow models, different from diffusion models, for text-to-3D generation. However, considering DDIM also conducts a deterministic sampling, and an image-noise pair can be coupled, computing the noise inversion is also effective to diffusion model. For example, DDIM inversion can also be used to resolve the over-smoothing and unrealistic outputs with SDS via diffusion models [NewRef-1].

[NewRef-1] Lukoianov et al., Score Distillation via Reparametrized DDIM, NeurIPS'24.

**Questions:**

In addition to W1 & W2 above, the authors can also provide answers for the questions below:

Q1. The authors claimed that FlowDreamer shows faster convergence, but I can't find the detailed analysis in training efficiency except for Figure 7, which shows minor improvements in training speed. Could the authors provide more analysis on convergence speed?

---

> ### Author Response · Authors · 2024-11-20
> **To reviewer bjYV.**
>
> Dear reviewer bjYV.
>
> ### W1.
> Thank you for pointing this out.
> We have adjusted the comparison method. Please refer to the unified response to all reviewers. Experiments demonstrate that our results yield high-fidelity outputs with richer textual details compared to other baseline methods using the same SD3 prior.
>
> ### W2.
> Thank you for your citation suggestion.
> This is a concurrent work with ours and indeed a very interesting paper. However, it focuses on a different aspect compared to our work. We are more focused on identifying the cause of over-smoothing and found that the smoothing effect in the rectified flow is caused by randomly sampling Gaussian noise. We directly addressed this issue by replacing the noise. We will also include this paper in the related work section later and provide a discussion on it.
>
> ### Q1.
> Thank you for pointing this out.
> We provided more examples for clarification. Please refer to Fig.12 and Fig.13. Experiments indicate that our FlowDreamer converges faster than VFDS. Additionally, the reason for faster convergence is that the optimization remains on the same trajectory due to the push-backward method, which ensures that the original noise can be found each time. This allows optimization to focus on coupling noise with the image, resulting in the same trajectory. In contrast, VFDS randomly samples noise each time(ref to Fig.5 in origal paper), leading to multiple optimization trajectories and thereby slower convergence.
>
> I am glad to discuss further with you if you have any other concerns.

---

> > ### Comment · Reviewer_bjYV · 2024-11-25
> >
> > I appreciate the authors' responses. For my better understanding, could the authors clarify the follow-up questions below?
> >
> > > This is a concurrent work with ours and indeed a very interesting paper. However, it focuses on a different aspect compared to our work. We are more focused on identifying the cause of over-smoothing and found that the smoothing effect in the rectified flow is caused by randomly sampling Gaussian noise.
> >
> > Is the over-smoothing effect is tailored only to rectified flow, and not applied to the diffusion models? I think diffusion models would have the similar side effect by using random noise sampling for SDS, and the previous work, I mentioned, also target to resolve that issue in addition to inefficiency.
> >
> > > Additionally, the reason for faster convergence is that the optimization remains on the same trajectory due to the push-backward method, which ensures that the original noise can be found each time.
> >
> > If the sampling starts from a pure noise, we can assure that the trajectory is unique. But, I think, even though we use the same noise, the intermediate rendering results changes overtime. Considering Reviewer W25F's comment, the sampling trajectory is not always straight and the trajectory can be changed over iterations. If my understanding is wrong, please feel free to point out it.
> >
> > One addition question (not affect to score): Since the proposed uses a single noise, could the results be too sensitive according to the seed? For example, even with the same text prompt, random seed can determine the quality of results and the variance would be high. It's often observed in image generation model, showing much different images on the same prompt by random seed.

---

> > > ### Author Response · Authors · 2024-11-25
> > > **To reviewer bjYV.**
> > >
> > > Q:Is the over-smoothing effect is tailored only to rectified flow, and not applied to the diffusion models?
> > >
> > > Thank you for pointing this out. As you have noted, this issue is not exclusive to rectified flow. Similarly, we have also observed this problem and mentioned it in the abstract: “However, empirical findings indicate that VFDS still results in over-smoothing outcomes.”. But we analyze the underlying reasons for such failure from the perspective of ODE trajectories and we also explored the impacts of various sampling methods and NFE, which differs from the work you mentioned. We will also include this paper in the related work section later and provide a discussion on it.
> > >
> > > Q:But, I think, even though we use the same noise, the intermediate rendering results changes overtime.
> > >
> > > In our framework, the parameters of the 3D model are trainable, but the parameters of the Flow model are frozen. These frozen parameters ensure that the conversion process between noise and image remains fixed. Therefore, even though the intermediate rendering results change, we can always obtain their corresponding noises via push-forwad operation, which forms the couple (image, noise). Our method utilizes both the coupling and reversible properties of flow model hence it can ensure the consistency. In contrast, VFDS cannot leverage the reversible property and randomly samples noise, which generates different couples and results in different trajectory updates, failing to ensure the consistency.
> > >
> > > In fact, apart from the intermediate rendering results changing, different camera poses also result in different images. However, the consistency can still be ensured under the same reason as explained above.
> > >
> > > Q: Considering Reviewer W25F's comment, the sampling trajectory is not always straight and the trajectory can be changed over iterations.
> > > There seems to be some misunderstanding here. The process we mentioned above does not require the trajectory to be straight. However, straightness affects the optimization speed, so under Rectified Flow, the optimization would theoretically be faster.
> > >
> > > Q:One addition question (not affect to score): Since the proposed uses a single noise, could the results be too sensitive according to the seed?
> > >
> > > That will not happen. When we used the sampling method Euler for push-backward, we found that the generated 3D results remained relatively stable. This is because 3D generation is different from image generation. In this framework, 3D generation requires about 30 minutes of optimization for each view, which often reduces diversity. Diversity in SDS can often be controlled by using different sampling time schedule, but this is also quite limited. In our method, this is relatively advantageous. As shown in the paper, we use different sampling methods (Euler for push-backward) and different NFE steps to control the diversity of the generated results. (Please refer to Fig. 9 in the paper and Fig. 5 in the Appendix.)
> > >
> > > **We are glad to discuss further with you if you have any other concerns.**

---

> ### Comment · Reviewer_bjYV · 2024-11-25
>
> I appreciate the authors' reply for my better understanding and correcting some of my misunderstanding. However, after reading the responses, I lean to keep my current score instead of increasing it. However, I will not block if the other reviewers' positive opinions.
>
> > Thank you for pointing this out. As you have noted, this issue is not exclusive to rectified flow. Similarly, we have also observed this problem and mentioned it in the abstract: “However, empirical findings indicate that VFDS still results in over-smoothing outcomes.”. But we analyze the underlying reasons for such failure from the perspective of ODE trajectories and we also explored the impacts of various sampling methods and NFE, which differs from the work you mentioned. We will also include this paper in the related work section later and provide a discussion on it.
>
> I'm still thinking that the motivation lacks of rationales. Considering a rectified flow (RF) can be a general form of diffusion models, I think VFDS is an extended version of SDS with RF's terminology. Thus, it's natural to expect the VFDS does not resolve the over-smoothing issue. However, the authors strongly assume that VFDS should resolve the issue, but there empirical finding, which VFDS does not resolve the issue, is non-trivial. In addition, I think UCM itself is not novel, since DDIM-inversion can also revert an image into a noise with deterministic scores' trajectory.

---

> > ### Author Response · Authors · 2024-11-25
> > **To reviewer bjYV.**
> >
> > Thank you for pointing this out.
> >
> > Q: However, the authors strongly assume that VFDS should resolve the issue, but there empirical finding, which VFDS does not resolve the issue, is non-trivial.
> >
> > I believe there is some misunderstanding of our paper. We do not strongly assume in the paper that VFDS should resolve the over-smoothing issue. We integrate SDS with the rectified flow model, and empirical findings indicate that VFDS still results in over-smoothing outcomes.
> >
> > Q:In addition, I think UCM itself is not novel, since DDIM-inversion can also revert an image into a noise with deterministic scores' trajectory.
> >
> > DDIM can indeed revert an image into noise, but DDIM is essentially just a step-skipping extension of DDPM. Its design, unlike RF, where $t \epsilon + (1-t)x = x_t$, does not inherently make image $x$ and $\epsilon$ symmetric. In contrast, RF’s symmetric linear interpolation design naturally possesses a reversible property. Therefore, using UCM to find the noise is a very straightforward and effective strategy. DDIM is discrete, and DDIM inversion requires first finding \(x_0\) at each step and then using the diffusion forward process to obtain the next noisy latent \(x_{t-1}\). In contrast, push-backward only involves taking an Euler step at each iteration to find the corresponding noise. This process is entirely continuous, meaning that at each step, the distance and sampling method are entirely determined by your settings.
> >
> > We are glad to discuss further with you if you have any other concerns.

---

> > > ### Comment · Reviewer_bjYV · 2024-11-25
> > >
> > > > I believe there is some misunderstanding of our paper. We do not strongly assume in the paper that VFDS should resolve the over-smoothing issue. We integrate SDS with the rectified flow model, and empirical findings indicate that VFDS still results in over-smoothing outcomes.
> > >
> > > I do NOT think I misunderstood. The authors still emphasize that, VFDS "still" results in, also in the paper. It's individual issue to "extend" SDS's formulation to RF using its "nomenclature". Since VFDS will not necessarily resolve the oversmoothing issue, I think the empirical finding is the contribution, but it's naturally expected regardless of VFDS,
> > >
> > > > In contrast, RF’s symmetric linear interpolation design naturally possesses a reversible property.
> > >
> > > I think it's theoretically wrong. After training, the RF's sampling trajectory is not straight line. It's the assumption of forward process and each step's prediction, while we're using multiple sampling steps even for RF.
> > >
> > > I also agreed with the other reviewer's opinions which this paper uses many high-level conjectures without enough theoretical evidences or over-state the contribution of the originality. Thus, I will maintain my score.

---

> ### Author Response · Authors · 2024-11-25
> **To reviewer bjYV.**
>
> Q: I think it's theoretically wrong. After training, the RF's sampling trajectory is not straight line. It's the assumption of forward process and each step's prediction, while we're using multiple sampling steps even for RF.
>
> Thank you for pointing this out.
>
> In the original paper on Rectified Flow[1], Algorithm 1 mentions: "The forward and backward sampling are equally favored by the training algorithm, because the objective in (1) is time-symmetric in that it yields the equivalent problem if we exchange $X0$ and $X1$ and flip the sign of $v$."
>
> The time-symmetric design is unrelated to whether the trajectory is straight. It emphasizes that the problem of mapping noise $\epsilon$ to the image $x$ and the reverse process of mapping $x$ to $\epsilon$ are equivalent problems. However, the design of the DDIM trajectory is not time-symmetric, so its reversibility is relatively not as good as that of RF.
>
> **We believe our goal in this discussion is to enhance both parties' understanding of the topic. Therefore, We are glad to discuss further with you if you have any other concerns.**
>
> [1]Liu, Xingchao, Chengyue Gong, and Qiang Liu. "Flow straight and fast: Learning to generate and transfer data with rectified flow." arXiv preprint arXiv:2209.03003 (2022).

---

> ### Comment · Reviewer_bjYV · 2024-11-25
>
> Thank you for explaining the reference. The property of "Time reversibility" in ODE itself is not the point, but both RF and diffusion models both can be reversible and it requires discretization error (note that diffusion model can also be time-continuous), but I still think that the authors try to claim this property is special only for RF. The point is that solving ODE also has a discretization error as diffusion model, which can also be continuous-time, and the inversion of diffusion models can also provide good results if we spend much computations for precise reversion in time. Since it's mentioned also in the other reviewer's comment, it's insufficient to explain a reference-based conjecture, but should be analyzed and shown directly in the paper (e.g. experiments and comparison).

---

> > ### Author Response · Authors · 2024-11-26
> > **To reviewer bjYV.**
> >
> > Thank you for pointing this out.
> >
> > You are right, ODE solvers inherently have reversibility. We started from the RF perspective, so we have conducted much of our analysis directly from that perspective. Diffusion models and RF have many similarities, but we believe this does not hinder exploring new ideas within the RF framework (and we believe our exploration is meaningful. We also greatly appreciate your suggestions, which have helped make our exploration more rigorous. Thank you.). In Fig. 30, we provide a comparison of the differences between the Diffusion model and RF when applying our method.
> >
> > **We still believe our goal in this discussion is to enhance both parties' understanding of the topic. Therefore, We are glad to discuss further with you if you have any other concerns.**

---

### Author Response · Authors · 2024-11-20
**Unified Response to All Reviewers.**

# Dear All Reviewers

Thank you to all the reviewers for your valuable feedback.
We have uploaded a new PDF, supplementing extensive experimental content on pages 11 to 29 of the paper. Please refer to these sections.
## The Issue of Unfair Comparisons

(1)In the baseline methods, SD2.1 was chosen, whereas Flow-based SD3 was used in our method. Taking the reviewers' suggestions into account, we replaced SD2.1 in the comparative methods with SD3, as shown in Fig. 10 and Fig. 11. However, since the baseline methods were specifically designed for the Diffusion model or adjusted 3D model parameters within Diffusion model, directly transferring them to SD3 results in **limited improvements and, in some cases, even worse performance** (for example, the DreamGaussian results for the prompt "an origami pig"). **Forcing a direct transfer to SD3 for comparison also leads to unfairness.**

(2) Therefore, we continued the examples shown in Figure 7 of the original paper and transferred the loss functions into a unified framework. In the original paper, we had already provided VFDS and VF-ISM; now, we have further transferred the Consistent3D loss into the unified framework, referred to as VF-CSD below. However, the original paper did not include many examples or experiments. To address this, we have conducted comparative experiments for a variety of prompts. All experiments use SD3 as the prior, with **the same random seeds, NeRF settings, and 3D GS settings; the only difference lies in the loss design.** We have provided a large number of experimental images for the reviewers to compare. Please refer to Fig. 17 to Fig. 29. The results with orange borders correspond to 3D GS, while the results with green borders correspond to NeRF. After comparison, **Our results yield high-fidelity outputs with richer textual details compared to other baseline methods using the same SD3 prior.**


## Comparing UCM and ISM

Many reviewers have also mentioned the comparison between UCM and ISM.
We created a table to better distinguish the differences between the two methods.

| **Method** | **ISM** | **UCM** |
| --- | --- | --- |
| **3D Model** | 3D GS | 3D GS and NeRF |
| **Prior** | Diffusion Model | Rectified Flow Model |
| **Planned Optimization loss** | Eq.(1) | ucm loss |
| **Actual Optimization loss** | Eq.(3) | ucm loss |
| **Implementation Method** | DDIM Inversion (discrete) | Push-backward (continuous) |
| **NFE** | 6 (5+1) steps | 4 (3+1) steps |


$\nabla_\theta \mathcal{L}{\text{SDS}}(\theta) = \mathbb{E}{t, \epsilon, c} \left[ \frac{\omega(t)}{\gamma(t)} \left( x_0 - \hat{x}_0^t \right) \frac{\partial g(\theta, c)}{\partial \theta} \right] (1)$

$\nabla_\theta \mathcal{L}(\theta) = \mathbb{E}{t,c} \left[ \frac{\omega(t)}{\gamma(t)} \left( \gamma(t) \left[ \epsilon\phi(x_t, t, y) - \epsilon_\phi(x_s, s, \varnothing) \right] + \eta_t \right) \frac{\partial g(\theta, c)}{\partial \theta} \right] (2)$

$\nabla_\theta \mathcal{L}{\text{ISM}}(\theta) := \mathbb{E}{t,c} \left[ \omega(t) \left( \epsilon_\phi(x_t, t, y) - \epsilon_\phi(x_s, s, \varnothing) \right) \frac{\partial g(\theta, c)}{\partial \theta} \right] (3)$


1. In terms of the 3D model, we selected 3D GS and NeRF and conducted extensive experiments on these two types of 3D models.

2. For the guidance prior, we directly adopted the state-of-the-art Rectified model.

3. The initial optimization function of LucidDreamer was Eq.(1) and through formula derivation, it led to Eq.(2). Then, by removing $\eta_t$ terms related to save time, they directly simplified it to Eq.(3). However, from the LucidDreamer paper, it is evident that the terms in $\eta_t$ are all of the same order of magnitude, which contains some unreasonable aspects.

4. Due to the advantages of the Rectified model, our push-backward method can be continuous, meaning the sampling method can be diverse.

5. Our NFE is smaller, resulting in less time for optimization.

6. The Step Size refers to the length traversed by DDIM inversion and Push-backward. Since our step size is 1, UCM aims to directly find a couple noise, whereas ISM inverse to the position of $x_t$ and proceed with further optimization.

---

### Author Response · Authors · 2024-11-23
**Quantitative comparisons on CLIP  similarity**

Quantitative comparisons on CLIP  similarity  in NeRF generation:
| Clip Model    | ViT-B-32 | ViT-L-14 | ViT-g-14 |
|---------------|----------|----------|----------|
| VFDS          | 32.13    | 31.85    | 31.78    |
| VF-CSD        | 32.46    | 31.57    | 32.02    |
| VF-ISM        | 32.72    | 32.96    | 33.14    |
| FlowDreamer   | **34.96**    | **34.19**    | **34.58**    |

Quantitative comparisons on CLIP  similarity  in 3D GS generation:
| Clip Model   | ViT-B-32 | ViT-L-14 | ViT-g-14 |
|--------------|----------|----------|----------|
| VFDS         | 28.32    | 28.48    | 29.08    |
| VF-CSD       | 28.36    | 28.03    | 28.56    |
| VF-ISM       | 29.56    | 29.52    | 29.87    |
| FlowDreamer  | **30.70**    | **30.49**    | **30.66**    |


**All experiments use SD3 as the prior, with the same random seeds, NeRF settings, and 3D GS settings; the only difference lies in the loss design.** To provide a clearer and more robust assessment, we randomly select 12 images with the same viewpoints from the rendered images. We employ three CLIP models from OpenCLIP, ViT-B-32, ViT-L-14, and ViT-g-14—to calculate the CLIP similarity. **Our FlowDreamer achieves superior CLIP similarity in both NeRF and 3D GS scenarios.**

---

### Author Response · Authors · 2024-11-28
**Unified Response to All Reviewers.**

Dear All Reviewers

 We sincerely thank all the reviewers for their valuable comments. Regarding the points raised by the reviewers: (1) whether RF is completely a straight line, (2) a comparison of convergence speed, (3) the fairness of the comparison, (4) additional ablation studies, (5) more experimental results, and (6) a more in-depth comparison with ISM, we have addressed all these points in both the main paper and supplementary materials. The reviewers can refer to the latest version of the PDF and supplementary materials.**The original paper has been made more rigorous, and the additional experiments provided in the supplementary materials are more comprehensive.**

**We believe that the goal of this discussion is to enhance the mutual understanding of the topic. Therefore, we are happy to discuss further if you have any other concerns.**

---

### Meta-Review · Area_Chair_Dgmn · 2024-12-19

**Metareview:**

This work introduces FlowDreamer, a novel approach that leverages pretrained text-to-image (T2I) models within the rectified flow framework for score distillation sampling (SDS). After detailing the training objective of SDS with a flow model, referred to as VFDS, the paper identifies the primary cause of underperformance as the use of multiple noises during optimization. To address this, FlowDreamer introduces a two-step process: first, it employs a flow model to perform rendered image inversion to recover the corresponding noise, and second, it applies VFDS. The proposed method demonstrates improved results compared to existing approaches and VFDS in text-to-3D generation tasks.

However, the majority of reviewers reached a consensus to reject this work. Key concerns include unclear contributions, unfair comparisons, limited novelty, missing ablation studies on straight trajectories, and various other experimental shortcomings. Consequently, the current version does not appear ready for publication. We strongly encourage the authors to address these concerns comprehensively, incorporating the reviewers' feedback to enhance the clarity, rigor, and impact of the work.

**Additional Comments On Reviewer Discussion:**

I mainly list the key concerns since different reviewers have different concerns.

1)	unclear contributions (reviwer bjYV, 3jWE)
The authors provide some explanations which but are not so convincing.

2)	unfair comparison (Reviewers bjYV, CLaj, pQyS).
The authors provide experimental results, and explain the reason why they use SD3, while others use SD2.1.

3)	insufficient novelty (reviwer 3jWE, CLaj, pQyS )
The authors do not provide extra experimental results, and only claim their target and contribution.

4)	various other experimental shortcomings (reviwer 3jWE, W25F, CLaj)
The authors provide some explanations which but are not so convincing.

Overall, I agree with the reviewers for most of the concerns.

---

### Decision · Program_Chairs · 2025-01-22

Reject